Citation: *Molecular Systems Biology* 9:672
www.molecularsystemsbiology.com

# The functional interactome landscape of the human histone deacetylase family

Preeti Joshi[1,3], Todd M Greco[1,3], Amanda J Guise[1], Yang Luo[1], Fang Yu[1], Alexey I Nesvizhskii[2] and Ileana M Cristea[1,*]

[1] Department of Molecular Biology, Princeton University, Princeton, NJ, USA and [2] Department of Pathology, University of Michigan Medical School, Ann Arbor, MI, USA
[3] These authors contributed equally to this work.
* Corresponding author. 210 Lewis Thomas Laboratory, Department of Molecular Biology, Princeton University, Washington Road, Princeton, NJ 08544, USA.
Tel.: +1 609 258 9417; Fax: +1 609 258 4575; E-mail: icristea@princeton.edu

Histone deacetylases (HDACs) are a diverse family of essential transcriptional regulatory enzymes, that function through the spatial and temporal recruitment of protein complexes. As the composition and regulation of HDAC complexes are only partially characterized, we built the first global protein interaction network for all 11 human HDACs in T cells. Integrating fluorescence microscopy, immunoaffinity purifications, quantitative mass spectrometry, and bioinformatics, we identified over 200 unreported interactions for both well-characterized and lesser-studied HDACs, a subset of which were validated by orthogonal approaches. We establish HDAC11 as a member of the survival of motor neuron complex and pinpoint a functional role in mRNA splicing. We designed a complementary label-free and metabolic-labeling mass spectrometry-based proteomics strategy for profiling interaction stability among different HDAC classes, revealing that HDAC1 interactions within chromatin-remodeling complexes are largely stable, while transcription factors preferentially exist in rapid equilibrium. Overall, this study represents a valuable resource for investigating HDAC functions in health and disease, encompassing emerging themes of HDAC regulation in cell cycle and RNA processing and a deeper functional understanding of HDAC complex stability.
*Molecular Systems Biology* **9**:672; published online 11 June 2013; doi:10.1038/msb.2013.26
*Subject Categories:* proteomics; chromatin & transcription
*Keywords:* HDAC; I-DIRT; interactions; proteomics; SAINT

## Introduction

The 11 human histone deacetylases (HDACs) are essential epigenetic regulators of gene transcription. HDACs act as components of multiprotein complexes, modulating transcription by removing acetyl groups from substrate lysines (Inoue and Fujimoto, 1969; Yang and Seto, 2008). By promotion of closed chromatin conformations and disruption of transcription factor activities, HDACs act in a finely-tuned balance with histone acetyltransferases to regulate transcription of downstream genes (Berger, 2007). Not surprisingly, HDAC dysfunction contributes to the progression of numerous human disease states, including cancers (Yang and Gregoire, 2005), viral infection (Murphy *et al*, 2002), cardiac disease (Bossuyt *et al*, 2008), and epigenetic response to drugs (Renthal *et al*, 2007).

Small molecules inhibiting HDAC activity are currently used in clinical trials for treatment of several cancers, including cutaneous T-cell lymphomas. Yet, HDAC inhibition is often marked by high cytotoxicity due to the requirement of HDAC activity in numerous cellular processes (Minucci and Pelicci, 2006). Moreover, the ability of HDACs to operate within many distinct complexes makes inhibitors that target a single HDAC a detriment to numerous cellular pathways. Therefore, the discovery of more selective targets, such as unique HDAC sub-complexes, is critical for the future design of single or combinatorial therapeutics. To achieve this level of selectivity, we require a better understanding of the ensemble of common and distinct HDAC interactions.

While the interactions and functions of many of the 11 human HDACs are not yet fully understood, it is well recognized that HDACs serve as scaffolds for a wide variety of spatially and temporally regulated interactions. The functions of HDAC-containing complexes are best understood for the class I enzymes, HDAC1 and HDAC2, which together with the histone-binding proteins, RBBP4 and RBBP7, form the core deacetylase complex. This functional unit is an essential component of chromatin-remodeling complexes, including the co-repressor of RE1 silencing transcription factor (CoREST), mSin3, and nucleosome remodeling and deacetylase (NuRD) (Tong *et al*, 1998; Wade *et al*, 1998; Xue *et al*, 1998; Zhang *et al*, 1998; You *et al*, 2001). The distinct functions of HDAC complexes are further illustrated by their involvement in viral infection. The NuRD complex is specifically recruited by the

human cytomegalovirus protein pUL38 to stimulate the activity of the viral major immediate-early promoter, playing a critical role in initiating infection (Terhune *et al*, 2010).

The spatial–temporal regulation of HDAC interactions is best exemplified by class IIa enzymes (HDACs 4, 5, 7, and 9). These HDACs have a unique localization-dependent mechanism of transcriptional control that is contingent on their nucleocytoplasmic shuttling (Miska *et al*, 1999; Grozinger and Schreiber, 2000; Wang *et al*, 2000; Kao *et al*, 2001). While their cytoplasmic interactions are not yet fully characterized, class IIa HDACs are known to repress transcription by interacting with transcription factors and co-repressor complexes in the nucleus (Wu *et al*, 2001; Fischle *et al*, 2002). Therefore, this shuttling acts as an effective spatial regulator of transcriptional repressive functions. In turn, nucleocytoplasmic shuttling itself is regulated by protein interactions and phosphorylation (Grozinger and Schreiber, 2000; McKinsey *et al*, 2000a; Zhao *et al*, 2001; Paroni *et al*, 2007; Greco *et al*, 2011). Recently, the number of identified phosphorylations on class II HDACs has seen a significant increase, yet the functional consequences of only a subset of these have been investigated (Ha *et al*, 2008; Greco *et al*, 2011; Guise *et al*, 2012). Highlighting the importance of phosphorylation- and interaction-dependent regulatory mechanisms of HDACs, increased nuclear export of class IIa HDACs is associated with dilated cardiomyopathy (Calalb *et al*, 2009).

Given their impact on human disease, selected HDACs have been the subject of intense study. Recent efforts have focused on understanding HDAC-regulated pathways in T cells during development and under pathophysiological conditions. T-cell-specific knockout of HDAC1 and HDAC2 in mice triggered thymocyte developmental arrest and misregulation of nearly 900 genes, including components of the T cell receptor signaling pathway (Dovey *et al*, 2013). As HDACs predominantly function within multiprotein complexes, the characterization of unique and shared protein complexes among the HDAC family members in T cells will provide insight into their cellular roles and potential targets for continued therapeutic development. However, the interactions and functions of many HDACs still remain unknown, and a systematic study of protein interactions across the entire HDAC family is lacking.

Here, we performed the first global proteomics and bioinformatics study of interactions of the 11 members of the human HDAC family. We established 11 separate CEM T-lymphoblast cell lines stably expressing functional EGFP-tagged HDACs as tools for proteomics and functional studies. We used proteomic and computational approaches for isolating and identifying HDAC-containing protein complexes. Through optimization of the label-free-based SAINT computational framework, we improved detection of specific interactions by accommodating larger dynamic ranges of spectral abundances. These identified interactions encompassed established chromatin remodeling complexes, such as Sin3, NuRD, and CoREST, as well as over 200 previously unreported HDAC interactions. We establish a previously unreported link between HDAC11 and survival of motor neurons (SMN)-containing complex with an essential role in spliceosomal snRNP assembly. We further demonstrate that downregulation of HDAC11 in T cells causes a functional U12-type splicing defect, resulting in mis-splicing of the ATXN10 gene.

In addition, we designed a hybrid, label-free and isotope-labeled, affinity purification approach to profile-relative interaction stability across the HDAC family members. This approach identified previously unreported stable HDAC interactions, and globally demonstrated that well-established chromatin remodeling HDAC1 interactions are largely stable within their complexes, while transcription factors preferentially exist in rapid equilibrium. Overall, by employing both global proteomic and targeted functional studies, we provide unique insights into less well-studied HDACs, a deeper functional understanding of HDAC complex stabilities, and a useful resource for investigating HDAC functions in health and disease states.

# Results

## Establishing functional EGFP-tagged histone deacetylases as tools for global interactome studies

The primary focus of this study was to build the first comprehensive network of functional protein interactions for all eleven HDAC enzymes. In part, the diverse functions of human HDACs are reflected by their division into sub-classes based on sequence homologies to the yeast deacetylases Rpd3 (class I) and Hda1 (class II): class I (HDACs 1, 2, 3, and 8); class IIa (HDACs 4, 5, 7, and 9) and IIb (HDACs 6 and 10); and class IV (HDAC11) (Grozinger *et al*, 1999; Cress and Seto, 2000; Verdin *et al*, 2003; Gregoretti *et al*, 2004). Common to all human HDACs is the presence of at least one core catalytic deacetylation domain required for enzymatic activities (Figure 1).

Given the use of T cells for HDAC inhibitor therapies and the importance of HDAC activity in T cell responses to immune and infectious diseases (Akimova *et al*, 2012), we sought to characterize HDAC interactions in this cell type. Using retroviral transduction, we generated CEM T-cell lines stably expressing each of the human HDACs (1–11) C-terminally tagged with both EGFP and FLAG (Figure 1). Next, we confirmed that the HDAC fusion proteins retained deacetylase activity using *in vitro* deacetylation assays with immunoaffinity purified GFP-tagged HDACs and an acetylated lysine substrate (Figure 1). To confirm that the EGFP tag does not interfere with subcellular localization, we examined each tagged HDAC by immunofluorescence microscopy. Expression of the EGFP tag alone demonstrates that EGFP displays a diffuse localization to both the nucleus and the cytoplasm in CEM T cells (Figure 1 and Supplementary Figure S1A), similar to our previous observations in EGFP HEK293 cell lines (Supplementary Figure S1A). Class I HDACs have been reported to localize to the nucleus, which was mimicked in our observations of EGFP-tagged HDAC localizations. Class IIa enzymes are known to shuttle between the nucleus and the cytoplasm, allowing these HDACs to interact with both nuclear and cytoplasmic proteins in a localization-dependent manner (Grozinger and Schreiber, 2000; McKinsey *et al*, 2000a; Zhao *et al*, 2001; Paroni *et al*, 2007; Greco *et al*, 2011). Consistent with their known shuttling ability, the EGFP-tagged HDAC4, 5, 7, and 9 were distributed to both nuclear and cytoplasmic compartments, with HDAC4 showing increased cytoplasmic

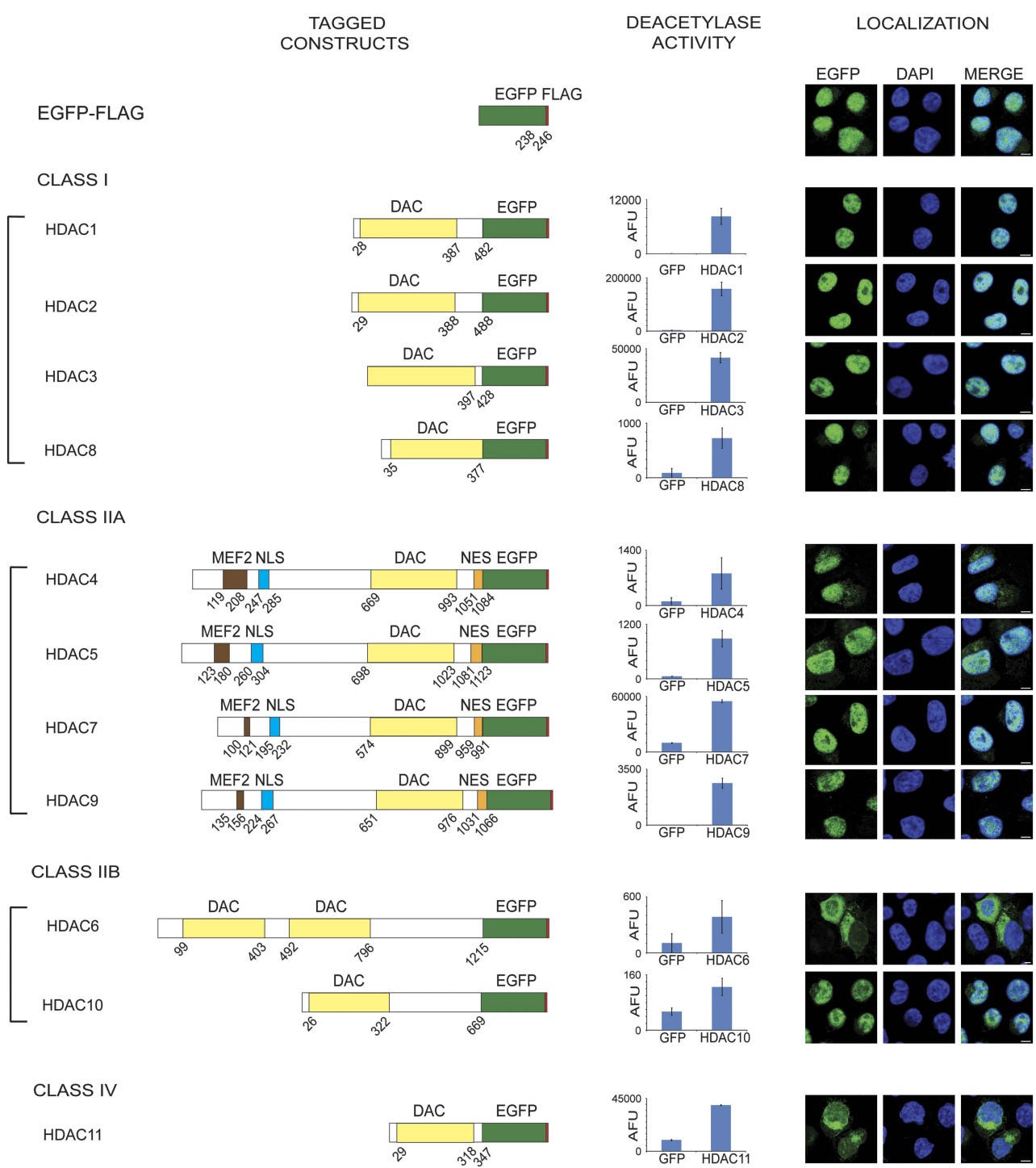

**Figure 1** Construction and validation of EGFP–FLAG tagged HDACs 1–11. (Left) HDAC 1–11 tagged with EGFP (green)–FLAG (red) at their C terminus. Boundaries of the deacetylase (yellow), MEF2 binding (brown), nuclear localization signal NLS (blue) and nuclear export signal NES (orange) regions are indicated. (Center) Deacetylase activity of HDACs isolated from CEM T cells measured using the Fluor-de-Lys assay ($n = 3$, AFU ± s.d.), as compared to EGFP–FLAG controls. (Right) Localization of EGFP–FLAG tagged HDACs in CEM T-cell lines using anti-GFP antibody (green); DNA is indicated by DAPI (blue); × 63 oil immersion lens; scale bar, 5 μm.

localization when compared to the remaining class IIa HDACs, which were instead predominately nuclear and only partially cytoplasmic. A similar dual localization phenotype was observed for the class IIb enzyme HDAC10, while HDAC6 was predominately cytoplasmic (Figure 1), consistent with previous reports of HDAC6 as a cytoplasmic deacetylase (Hubbert *et al*, 2002). While the morphology of CEM T cells grown in suspension provides a minor challenge in visualizing

cytoplasmic proteins, as the nucleus occupies a substantial fraction of the total cell volume, close examination illustrates the HDAC6 cytoplasmic enrichment and the dual localizations detailed above (Supplementary Figure S1B). Therefore, while HDAC localizations have not previously been fully characterized in T cells, our results agree with endogenous protein localizations reported for HDACs in various cell types (Yang and Seto, 2008; Keedy *et al*, 2009).

In contrast, studies of HDAC11, the sole class IV HDAC, have reported divergent localization patterns in different cellular models. Transiently transfected epitope-tagged HDAC11 in HEK293 cells was found to be predominantly nuclear (Gao *et al*, 2002), while HDAC11 was observed in the cytoplasm of resting CD4+ T cells (Keedy *et al*, 2009). Given this possible cell type-dependent localization, we evaluated the localizations of endogenous and EGFP-tagged HDAC11 in CEM T cells by microscopy. HDAC11–EGFP was localized to both cytoplasmic and nuclear compartments (Figure 1). A similar localization was observed for HDAC11 in wild-type CEM T cells using two separate antibodies raised against the endogenous protein (Supplementary Figure S2). Unique among HDACs, the cytoplasmic distribution of HDAC11–EGFP and endogenous HDAC11 often appeared concentrated asymmetrically in the perinuclear region.

Collectively, confocal microscopy analyses and *in vitro* activity assays demonstrated that the C-terminally EGFP-tagged HDACs maintain both enzymatic activity and wild-type-like localizations in CEM T cells. Therefore, these cellular models constitute functionally relevant tools for studying the interactome of HDACs.

## Efficient immunoaffinity purifications of the 11 human histone deacetylases

We next developed an optimized workflow using affinity purification coupled to mass spectrometry-based proteomics (AP-MS) (Miteva *et al*, 2013) to characterize the protein interaction profiles of all HDACs. This workflow employed (1) cryogenic cell lysis and rapid immunoaffinity purification of the bait protein (HDAC–EGFP) via antibody-coupled magnetic beads, (2) complementary proteomics–bioinformatics analyses to identify co-isolated proteins, assess specificity and stability of interaction, and determine functional relationships, and (3) targeted studies to confirm putative HDAC interactions (Figure 2A). Isolations were optimized for efficient recovery of each HDAC. Cryogenic cell lysis maximized disruption of subcellular and cytoskeletal structures, increasing HDAC recovery from nuclear and cytoplasmic compartments, while reducing non-specific associations. Due to the differential localizations, expression levels, and biophysical characteristics of different HDACs, various lysis buffer compositions were assessed for efficiency of HDAC–EGFP solubilization and isolation. The optimized lysis buffers (Supplementary Table S1) afforded efficient isolation of each EGFP-tagged HDAC, as confirmed by western blot (Figure 2B). Additionally, Coomassie staining of immunoisolates separated by SDS–PAGE confirmed HDAC isolation, as indicated by representative gel lanes for all 11 HDACs compared to an EGFP control (Figure 2B).

## Adapting SAINT to assess interaction specificity within the heterogeneous HDAC interactome

For global comparison of protein interaction profiles among EGFP-tagged HDACs, we first employed a label-free affinity purification strategy coupled to 1D-nanoliquid chromatography-tandem mass spectrometry (1D-nLC-MS/MS). Interaction specificity was assessed by the SAINT (Significance Analysis of INTeractions) algorithm using nLC-MS/MS spectral count data. SAINT statistically models spectral counts between controls (EGFP) and tagged bait samples (HDAC–EGFP) to calculate probabilities of interaction specificity (Choi *et al*, 2011). Analyses were performed for each EGFP-tagged HDAC ($N = 2$–3 biological replicates) versus the EGFP controls ($N = 7$ biological replicates) (Supplementary Table S2). Initially, this interactome data set for all HDACs presented several challenges in determining interaction specificity, as the isolates had significant differences in the dynamic range of spectral counts and heterogeneity of co-isolated proteins among different HDAC baits. In particular, HDAC1 and HDAC2 yielded greater total numbers of identified prey proteins and, on average, higher spectral counts for each prey protein. This is consistent with the BioGRID database (Stark *et al*, 2006), which, among all HDACs, has the highest number of known interactions for HDAC1. To address this, several aspects of SAINT were optimized. Rather than use a single model across all data sets, SAINT modeling was performed individually for each HDAC bait compared to the control data set. To minimize the negative effect of prey proteins with high spectral counts, each protein's spectral counts in the isolations were normalized by the ratio of average total spectral counts in controls versus the respective HDAC sample. Selection of SAINT score thresholds were aided by generating ROC-like curves for HDAC1, 3, and 4, for which the greatest number of HDAC interactions have been cataloged. We determined putative protein interactions at different SAINT score thresholds. Then, by comparison to the iRefIndex database (Turner *et al*, 2010), we approximated true-positive and false-positive rates based on presence or absence in the iRefIndex database, respectively (see Supplementary Figure S3). As we do not have reliable estimates of error rates for the lesser studied HDACs, our selection of initial SAINT score thresholds were conservative. We considered prey proteins with an average score of $\geqslant 0.75$ in at least one HDAC isolation as putative specific interactions, except for HDAC1/2 and HDAC11 preys, which required an increased stringency of $\geqslant 0.90$ (see Supplementary Figure S3A) and $> 0.95$, respectively. Although MEF2C, a well-known interaction among the class II HDACs (Lu *et al*, 2000), was identified in the HDAC immunoisolates, given its lower abundance it did not generate significant spectral counts to pass the SAINT score filters (Supplementary Table S2) and was therefore manually included in subsequent analyses. Yet, even at these relatively stringent thresholds, we identified between 40–60% of known interactions in the iRefIndex (Supplementary Figure S3A–C), which includes interactions identified in various cell types and experimental conditions.

After removal of non-specific interactions using SAINT (Supplementary Table S3), 281 proteins across all HDACs were considered as putative specific interactions (Supplementary Table S2). Due to the relative lack of reported HDAC11

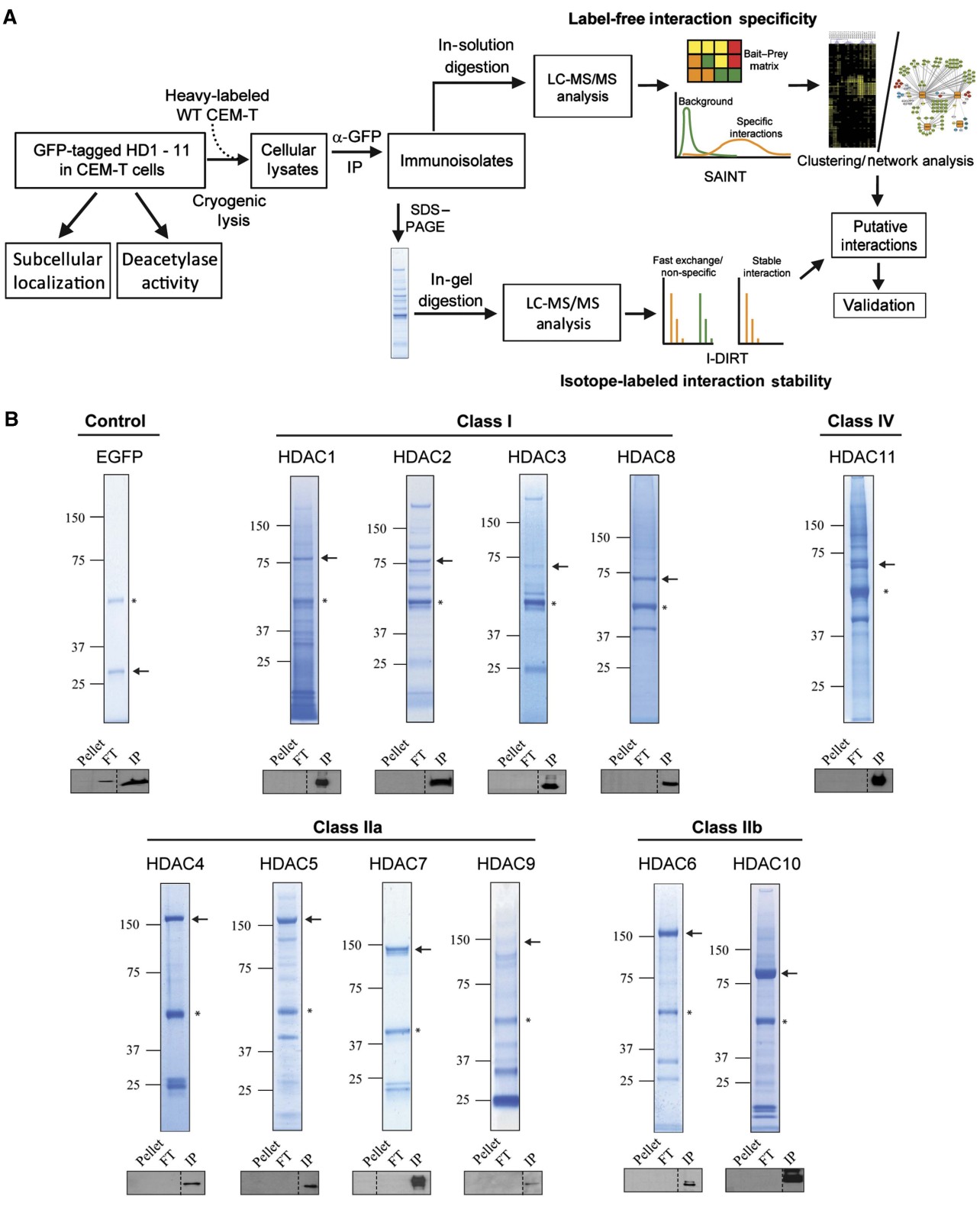

**Figure 2** Proteomic workflow and immunoisolation of 11 EGFP-tagged human HDACs. (**A**) Workflow for immunoisolation of HDACs from CEM T-cell lines stably expressing HDAC–EGFP. HDAC–EGFP immunoisolates were subjected to label-free and isotope-labeled AP-MS workflows using SAINT and/or I-DIRT analysis, respectively. Hierarchical clustering and interaction networks visualized HDAC–HDAC and HDAC–prey relationships. Candidate protein interactions from global AP-MS were supported by molecular imaging and biochemical approaches. (**B**) Representative SDS–PAGE separations of Coomassie-stained EGFP-tagged HDAC1–11 immunoisolates. EGFP only immunoisolate is shown as a control. Arrows indicate the band containing the isolated bait. *, contaminant band. Western blotting assessed efficiency of HDAC–EGFP recovery in elution (IP) fraction relative to unbound flowthrough (FT) and insoluble cell pellet (pellet). Ten percent of each fraction was analyzed.

interactions and the large number of SAINT-filtered proteins, we excluded HDAC11 from the global interaction map, performing an independent analysis of its interactions. Overall, this proteomics–bioinformatics approach using nLC-MS/MS paired with SAINT provided a transparent strategy for selection of scoring thresholds that excluded likely non-specific proteins from HDAC affinity purifications.

## Protein interaction clustering reflects phylogenetic relationships and functional commonalities among distinct HDACs

Using SAINT-filtered protein interactions, we next performed hierarchical clustering to profile prey proteins that are unique to specific HDAC classes or common across the HDAC interactome. In total, 180 interactions across HDACs 1–10 (23 independent isolations) were clustered by Pearson correlation. HDAC bait spectral counts were removed to prevent clustering bias, and $\log_2$-transformed prey spectral counts were used to calculate the distance matrix. Dendrograms were assembled in a heatmap as a function of HDACs and associated proteins (Figure 3). Importantly, clustering of biological replicates showed a high degree of similarity to each other, supporting our isolation reproducibility and selection of SAINT probability filters to remove non-specific interactions.

From a functional perspective, clusters formed among different HDACs often reflected their phylogenetic classifications. For instance, among the class I HDACs, HDAC1 and HDAC2 were assembled into a unique cluster, with their respective prey proteins forming the largest cluster of 83 genes (Figure 3, *red*). This is consistent with HDAC1 and HDAC2 comprising a catalytic core that functions as part of several multi-protein complexes, including NuRD (Tong *et al*, 1998) and CoREST (You *et al*, 2001). While many interactions in this cluster were exclusive to HDAC1 and HDAC2, commonalities with other HDACs were also highlighted. Specifically, members of the CoREST complex were also found to associate with HDAC3, namely, lysine-specific histone demethylase 1A (KDM1A/LSD1) (Bantscheff *et al*, 2011), as well as REST co-repressor 1 (RCOR1) and Ras-responsive element-binding protein 1 (RREB1), whose associations with HDACs have not been reported.

Three class IIa members, HDAC4, 5, and 7, were part of a single cluster. For this class, clustering was driven by shared interactions with the nuclear co-repressor complex (NCoR) (Figure 3, *orange*) and the 14-3-3 proteins (Figure 3, purple), which facilitate nucleocytoplasmic shuttling of class IIa HDACs (Grozinger and Schreiber, 2000; McKinsey *et al*, 2000b; Kao *et al*, 2001; Yang and Gregoire, 2005). In contrast, the class IIb enzymes, HDAC6 and HDAC10, were not part of the same cluster. Given their contrasting subcellular distributions, protein localization may be a stricter determinant of protein interactions than phylogenetic relationship. Interestingly, HDAC6, HDAC8, and HDAC9 were not part of larger clusters, each forming their own distinct gene cluster (Figure 3, *yellow, teal, and blue*). Overall, hierarchical clustering of prey proteins across the HDAC family highlighted several known, shared features among the class I and II HDACs. This provides strong support for the suitability of our HDAC–EGFP CEM T-cell model system for

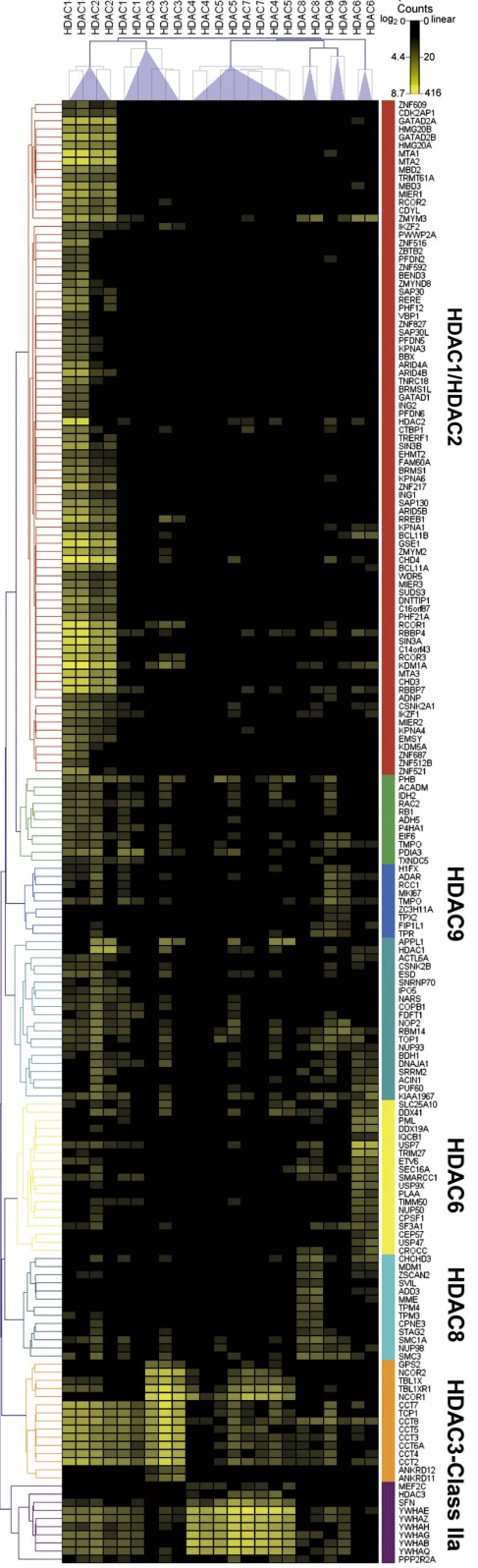

**Figure 3** Clustering of HDAC protein interaction profiles. Hierarchical clustering analysis of HDAC1–10 and 180 SAINT-filtered prey proteins. Clustering was performed as a function of $\log_2$-transformed spectral counts using Pearson correlation and average linkage between biological replicates from 23 independent HDAC–EGFP isolations. Prey clusters were color coded according to the respective dendrogram.

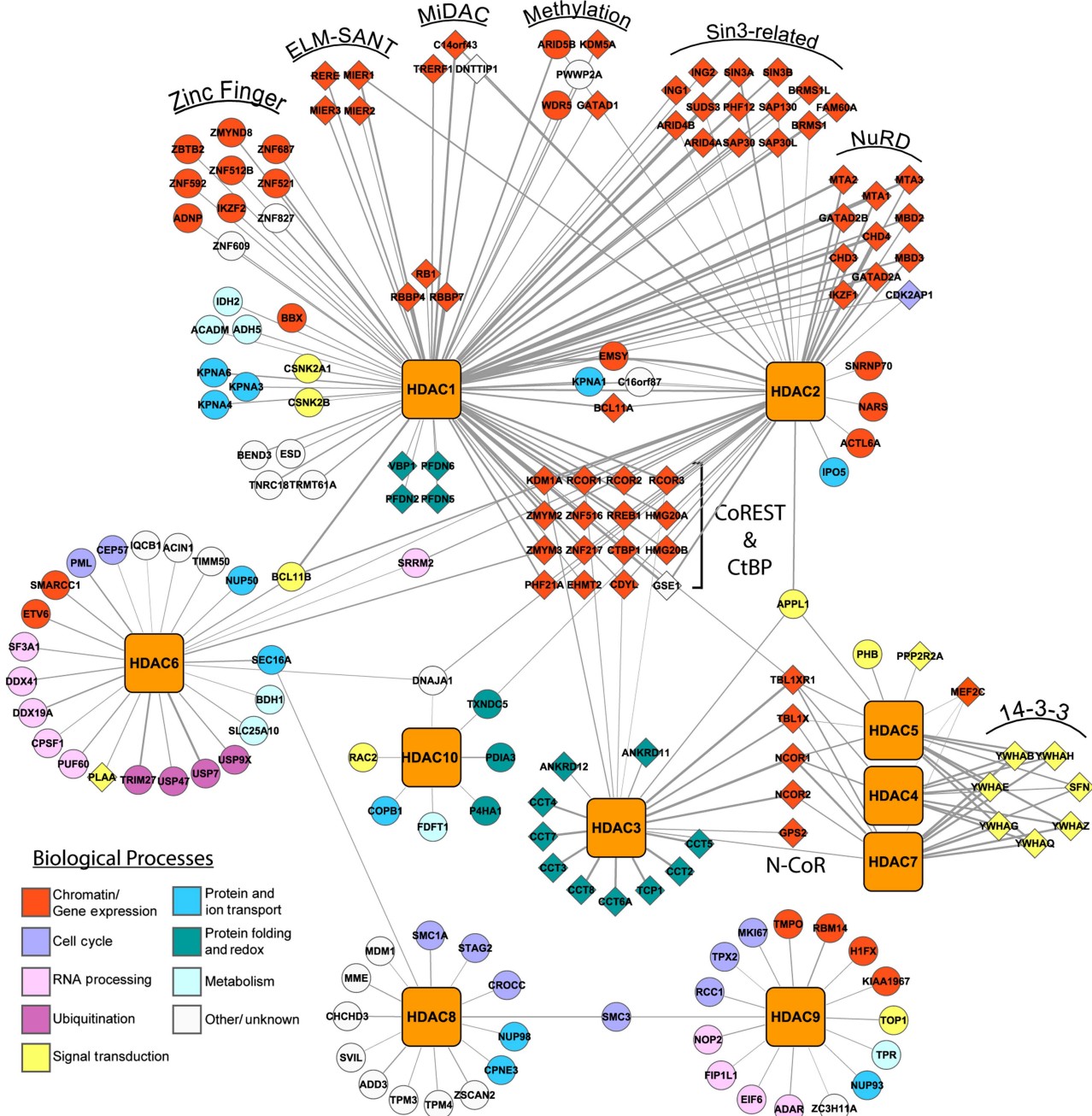

**Figure 4** Comprehensive HDAC interaction network. Cytoscape interaction network representing 180 SAINT-filtered putative HDAC–prey interactions. HDAC–prey interactions were visualized by network edges. Preys were manually classified and color coded by biological processes. If known, preys were also grouped by subcomplex or function. Diamond and circle nodes indicate previously identified and uncharacterized HDAC interactions, respectively. Edge thickness indicates $\log_2$-transformed prey spectral counts.

comparing HDAC associations. Moreover, this approach emphasizes the utility of evaluating HDACs by their interactions, in addition to their phylogenetic relationships.

## Building functional interaction networks of histone deacetylases

While hierarchical clustering provided an effective way to assess interaction profiles and unique gene clusters, it did

not directly convey the unique association of proteins with individual HDACs. Therefore, an HDAC-centric interaction network was constructed from the SAINT-filtered prey proteins (Figure 4). Importantly, for the well-characterized HDACs, such as HDAC1 and HDAC2, the majority of our specific interactions were in agreement with those annotated in the literature (Figure 4, *diamonds*). For example, all 11 members of the NuRD complex were co-isolated with HDAC1 and HDAC2. We specifically co-isolated with HDAC1 and/or HDAC2, the known members of the mitotic deacetylase complex (MiDAC),

TRERF1, DNTTIP1, and MIDEAS/C14orf43 (Bantscheff *et al*, 2011). As the MiDAC complex was previously detected during mitosis, its isolation from asynchronous cells supports the sensitivity of our methods to capture lower abundance interactions.

Notably, we identified 29 previously unreported putative HDAC1 interactions (Figure 4, *circles*), 11 of which are implicated in chromatin remodeling and gene expression (Figure 4, *red circles*). Also, 10 of these interactions are with zinc finger domain-containing proteins, which is of interest, as zinc finger proteins are known components of chromatin remodeling complexes, including CoREST (e.g., ZMYM2 and 3) and NuRD (e.g., IKZF1). HDAC1 also co-isolated WDR5, ARID5B, and PWWP2A, which have been shown either to directly regulate histone methylation or recruit demethylase complexes to histone-bound DNA (Han *et al*, 2006; Vermeulen *et al*, 2010; Baba *et al*, 2011).

Among the less well-characterized HDACs, most interactions were found to be unique to distinct HDACs. For example, most identified HDAC8 associations have not been reported and are largely uncharacterized (Figure 4, *white circles*). Intriguing was the HDAC8 interaction with components of the cohesin complex, SMC1A, SMC3, and STAG2, involved in sister chromatid segregation during mitosis (Barbero, 2009). HDAC8 was recently shown to deacetylate SMC3, with loss-of-function HDAC8 mutations showing impaired cohesin complex regulation and being linked to the congenital malformation disorder, Cornelia de Lange syndrome (Deardorff *et al*, 2012). Interestingly, our results show that HDAC9 can also associate with SMC3 (Figure 4). Another noteworthy HDAC9 interaction was KIAA1967, also known as deleted in breast cancer 1 (DBC1), which was implicated in inhibition of HDAC3 (Chini *et al*, 2010) and SIRT1, an $NAD^+$-dependent deacetylase (Kim *et al*, 2008; Zhao *et al*, 2008). Further illustrating the diverse cellular functions mediated by HDACs, HDAC6 was found to associate with proteins involved in ubiquitination, including USP7, USP47, USP9X, and TRIM27 (Figure 4). We also identified the reported interaction with phospholipase A-2-activating protein (Seigneurin-Berny *et al*, 2001), which directly binds ubiquitin (Fu *et al*, 2009), further strengthening HDAC6 roles in ubiquitin-dependent processes.

In summary, building an interaction network for human HDACs allowed us to distinguish highly interconnected interactions among multiple HDACs versus those unique to individual HDACs. This analysis highlighted unreported interactions for the HDAC1/HDAC2 core deacetylase complex, and importantly, for the less well-studied HDACs. Using functional protein classification, we provide evidence for HDACs in ubiquitination, cell cycle regulation, and define molecular targets to extend the understanding of HDAC regulation.

## HDAC11 associates with the SMN complex and regulates mRNA splicing

Based on the relative lack of knowledge regarding HDAC11 interactions ($<10$ interactions in BioGRID) and the large number of SAINT-filtered proteins identified in our isolations (Supplementary Table S2), we chose to independently assess

HDAC11 associations. Similar to our global analysis of HDACs 1–10, we subjected HDAC11 immunoisolations to a label-free workflow to filter non-specific interactions. In total, 124 prey proteins passed the SAINT probability threshold ($>0.95$). While the majority of observed HDAC11 interactions had nuclear gene ontology annotations, a significant number ($n = 32$) had a dual cytoplasmic/nuclear annotation (Figure 5A). This finding is interesting given the dual cellular localization of HDAC11 in T cells (Figure 1, Supplementary Figure S1). The predominant biological processes were related to chromatin modification/gene expression (Figure 5A, *red*) and, intriguingly, RNA editing and processing (Figure 5A, *pink*). Phylogenetic evidence suggests HDAC11 diverged from its ancestral gene(s) earlier in their evolution than class I and II HDACs (Gao *et al*, 2002); thus, HDAC11 may have acquired separate cellular roles and protein interaction-dependent functions. To explore this possibility, we utilized ClueGO (Bindea *et al*, 2009) to perform a gene ontology comparison between proteins co-isolated with HDAC11 ($n = 124$) and those isolated with other HDACs ($n = 180$, Supplementary Table S2). Relative to the entire interactome, HDAC11 interactions were significantly enriched in biological processes such as spliceosomal RNA processing and ribonucleoprotein complex biogenesis (Figure 5B, *pink/red*). In fact, proteins assigned to several RNA processing-related functions were exclusively found in HDAC11 isolations. In contrast, HDAC11 interactions were under-represented in the term 'NLS-bearing substrate import into the nucleus', a characteristic class II function (Figure 5B, *green*). The enriched functional attributes of HDAC11 interactions are consistent with the limited overlap of the putative protein interactions with the other HDACs (Supplementary Figures S4 and S5).

To assess whether proteins enriched with HDAC11 participate in common signaling pathways or cellular processes, STRING analysis was performed (Szklarczyk *et al*, 2011). While several distinct STRING networks were assembled (Supplementary Figure S2), given the prominence of the spliceosomal RNA processing ontology, we focused on the functional association between an SMN sub-network (SMN1, Gemin3, and Gemin4) (Charroux *et al*, 1999, 2000; Meister *et al*, 2000) and cohesin subunits (SMC1A, SMC3, and STAG1) (Peters *et al*, 2008), linked through nucleoporins (Doye and Hurt, 1997). Another interesting interaction in this network was the endoribonuclease enzyme Dicer1, which directly interacts with the nucleoporin NUP153 (Ando *et al*, 2011). We integrated mass spectrometry-based relative abundance measurements to predict proteins that may serve as key nodes through which HDAC11 could exert its functions (Figure 5C, Supplementary Table S4). As our lab has recently shown, estimating the relative abundance of interacting proteins by NSAF (normalized spectral abundance factor) values, (Zybailov *et al*, 2007) normalized by their PAX cellular abundance measurements (Wang *et al*, 2012) can reveal proteins or sub-complexes of significant interest within an existing functional network (Tsai *et al*, 2012). Interestingly, the SMN sub-network and cohesins had relatively high enrichment indices (Figure 5C, *node size*). Based on these data, we performed reciprocal isolations with antibodies against SMN1, Dicer1, Gemin3, and Gemin4. We found that HDAC11–EGFP was co-isolated with each endogenous protein (Figure 5D).

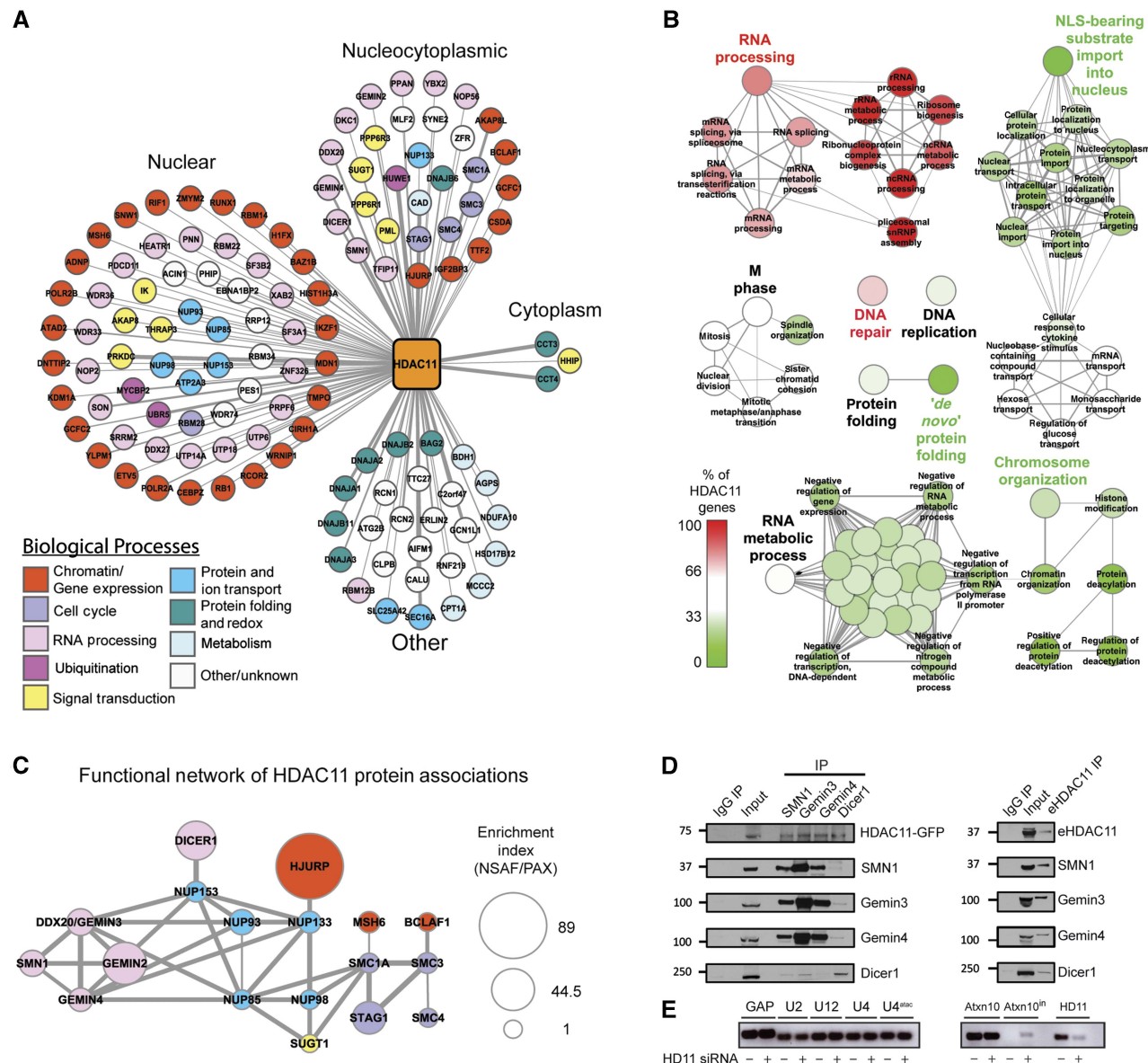

**Figure 5** HDAC11 functional interaction network analysis identifies components of snRNP biogenesis complexes. (**A**) Cytoscape interaction network of putative HDAC11 interactions. 124 SAINT-filtered proteins co-isolated with HDAC11 were grouped by subcellular localizations. Proteins were color coded by biological processes. Circle-shaped nodes indicate previously unreported interactions. (**B**) GO biological process (GO BP) network comparing classifications of HDAC11 versus the HDAC1–10 interactome data set. GO BP terms, assigned by the ClueGO Cytoscape plugin, depict functions that are (1) common (white circles, 33–66% of HDAC11 genes), (2) enriched in HDAC11 (red circles, > 66% of HDAC11 genes), or (3) enriched in the HDAC1–10 interactome (green circles, < 33% of HDAC11 genes). For clarity, a subset of GO BP term labels relating to detailed RNA metabolic processes were removed (see Supplementary Figure S6A). (**C**) STRING functional network of prominent candidate HDAC11 interactions visualized in Cytoscape. Nodes were color coded by biological processes indicated in (A). Node size was expressed as an enrichment index, which is the protein's normalized spectral abundance factor (NSAF) relative to its estimated cellular abundance in the PAX database, normalized to SUGT1 set at an arbitrary value of one. (**D**) Validation of selected HDAC11 interactions by reciprocal isolations and immunopurification of endogenous HDAC11 (eHDAC11). IgG was used as a control. Left, immunoaffinity purifications of endogenous SMN1, Gemin3, Gemin4, or Dicer1, and detection of complex members by western blot. Right, immunoaffinity purification of eHDAC11. (**E**) Splicing defects in the *ATXN10* U12-type intron 10 upon knockdown of HDAC11 in WT CEM T cells. RNA levels upon treatment with HDAC11 siRNA or a scrambled control were quantified by qRT–PCR (*n* = 3). A representative agarose gel is shown to visualize the levels of indicated PCR products. Left, mRNA of GAPDH (GAP) and snRNA of U2, U12, U4 and U4atac. Right, mRNA of *ATXN10*, *ATXN10*_intron, and *HDAC11*.

Additionally, isolation using an antibody against HDAC11 confirmed these interactions within endogenous HDAC11 complexes (Figure 5D).

The validated association of HDAC11 with the SMN complex prompted us to examine possible cellular roles of this HDAC-containing complex. Recent reports suggested that SMN1 deficiency correlates with decreases in levels of the minor U4atac/U6atac/U5 tri-snRNPs, promoting accumulation of mis-spliced U12-type introns (Campion *et al*, 2010; Boulisfane *et al*, 2011). Homozygous deletion of SMN1 was shown to be accompanied by intron retention in the U12-type intron from the *ATXN10* and *Thoc2* genes in lymphoblasts

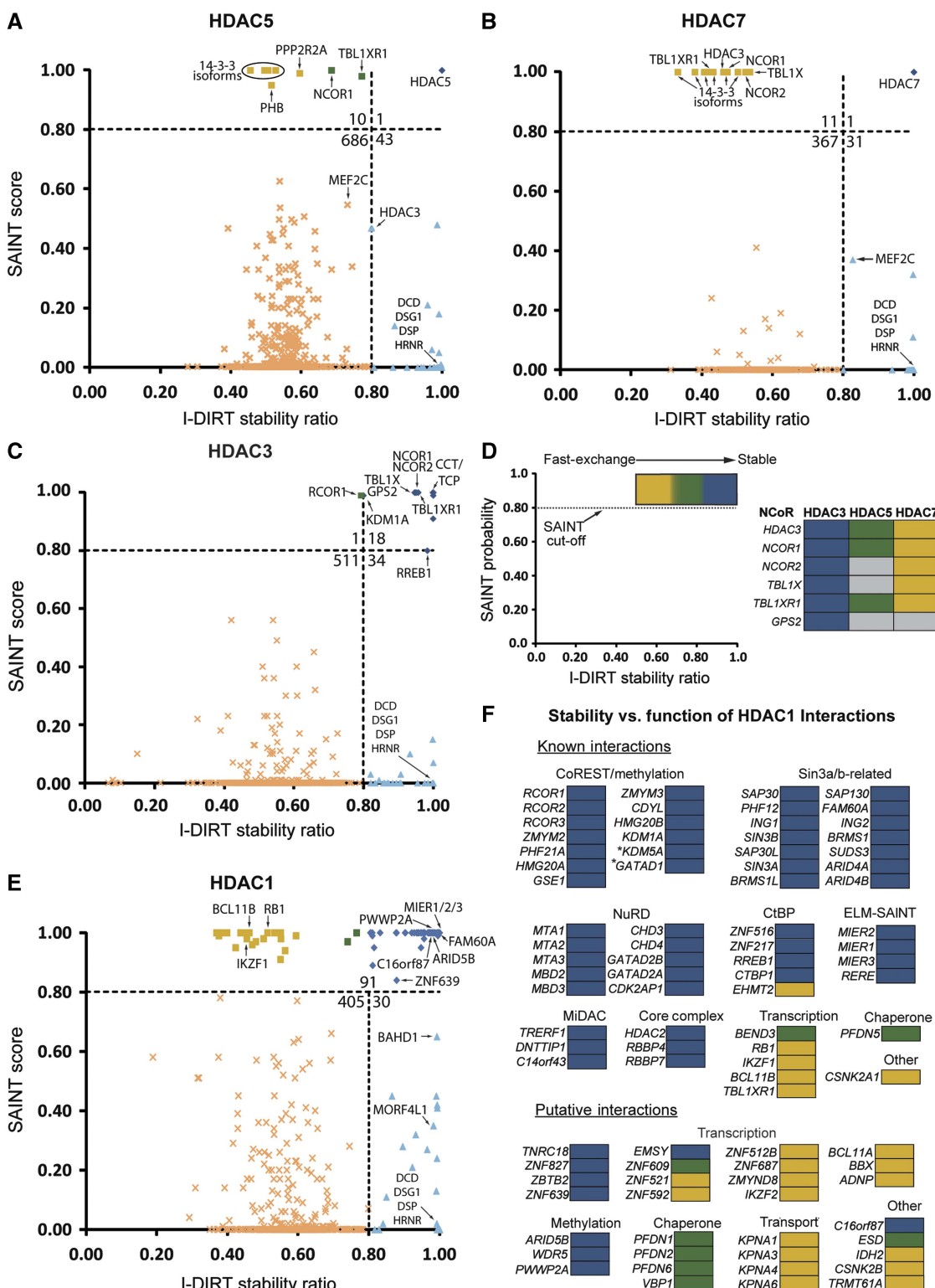

**Figure 6** Profiling of relative interaction stability within HDAC-containing complexes. Scatter plots show the relationship between interaction specificity (SAINT scores) and stability/specificity (I-DIRT ratios). Data shown are for common protein identifications between label-free and isotope-labeled AP-MS approaches from (**A**) HDAC5, (**B**) HDAC7, (**C**) HDAC3, and (**E**) HDAC1 isolations. Dashed lines represent selected thresholds and total protein number in each quadrant is shown. Selected data points are labeled with gene symbols. (**D**) Left, a region of high SAINT specificity, but varying I-DIRT ratios (0.5 to 1.0), is indicated by a color coded gradient indicating a stability range. Right, the relative stability of NCoR complex members is compared for HDAC3, 5, and 7. Gray boxes indicate the protein was absent or below SAINT threshold. (**F**) Known (top) and putative (bottom) HDAC1 interactions with SAINT scores > 0.80 (*n* = 90) are listed as gene symbols, depicted with their relative stability (D), and classified by known HDAC1 complexes or cellular function.

 

derived from patients with spinal muscular atrophy (Boulisfane *et al*, 2011). We therefore tested the hypothesis that HDAC11 downregulation would lead to similar splicing defects through disruption of spliceosome function. Upon knockdown of HDAC11 in wild-type CEM T cells, using qRT–PCR, we observed an accumulation of mis-spliced *ATXN10* mRNA, which was not detected in the non-targeted siRNA control (Figure 5E). A similar analysis of *Thoc2* found no splicing defect in the I37 intron from this gene (Supplementary Figure S6B). HDAC11 knockdown did not significantly affect the mRNA levels of the major snRNAs, as assessed by the levels of U2 and U4, or the minor snRNAs, U12 and U4atac. Overall, these results demonstrate that HDAC11 is involved in mRNA splicing and that it functionally associates with the SMN complex.

## An integrative approach for profiling protein interaction stabilities within isolated HDAC complexes

Given that HDACs are core components of complexes that are dynamically recruited for transcriptional regulation, the relative binding affinities of HDAC interactions can provide insight into the cellular function and temporal regulation of these complexes. To distinguish stable from fast-exchanging interactions, we designed an approach integrating label-free and metabolic-labeling methods. The label-free method using SAINT scoring, as described above, distinguishes specific from non-specific associations to the beads, antibody, and/or the EGFP tag, but does not provide information about the relative stability of interactions. Therefore, we integrated the metabolic-labeling I-DIRT approach (Tackett *et al*, 2005) into our proteomic workflow (Figure 2A). While originally designed to account for non-specific associations to the isolated protein complexes themselves, we tested whether the combination of I-DIRT and SAINT provide a measure of relative interaction stability for specific interactions. For metabolic labeling, wild-type CEM T cells were isotopically labeled by SILAC (Ong *et al*, 2003) and mixed prior to cryogenic lysis with an equal amount of CEM T cells stably expressing HDAC–EGFP cultured in 'light' media. Quantification of SILAC peptide pairs was performed, and median isotope protein ratios were expressed as the ratio of heavy abundance/total light and heavy abundance. Isotope ratios would approximate relative interaction stabilities with fast-exchanging interactions having ratios closer to 0.50 and increasingly stable interactions reaching a maximum ratio of 1.0.

We first established the feasibility of this approach to assess relative stability for interactions for HDAC5 and HDAC7, which undergo nucleocytoplasmic shuttling *in vivo*. Their nuclear export relieves HDAC-dependent transcriptional repression (McKinsey *et al*, 2000a), and is accompanied by loss of interaction with the NCoR complex (Greco *et al*, 2011). Therefore, we hypothesized that these class II interactions would be the most dynamic among HDAC sub-families and be identified as fast-exchanging partners by I-DIRT. As predicted, interactions identified as specific by SAINT for both HDAC5 and HDAC7 had I-DIRT ratios of < 0.80, suggesting these interactions are dynamic (Figures 6A and B). These proteins included members of the NCoR complex: NCOR1, NCOR2/

SMRT, TBL1X, TBL1XR1, and HDAC3, and the 14-3-3 chaperone proteins, which are intimately involved in shuttling. In fact, the majority of these well-established interactions were clustered at a ratio of ~ 0.50, suggesting that during complex isolation, their *in vitro* rates of exchange reached an equilibrium.

In contrast, isolation of HDAC3, a class I HDAC and itself a member of the NCoR complex, revealed a significantly different distribution, as nearly all interactions had > 0.80 I-DIRT ratios. In particular, ratios for the NCoR complex members were consistently greater in HDAC3 as compared to class II enzymes (Figures 6C and D). These data suggest that the relative stabilities of NCoR complex members are greater within HDAC3-containing complexes, as opposed to HDAC5- or HDAC7-containing complexes. This is consistent with studies supporting a direct HDAC3-NCoR interaction, but an indirect interaction of class II HDACs with NCoR via an HDAC3 bridge (Fischle *et al*, 2002). These results demonstrate the ability of our complementary approach to discriminate between stable and fast-exchanging interactions and further underscore the functional dichotomy between class I and II HDACs.

## Profiling stability of HDAC1 interactions reveals a functional segregation of transcription and multi-protein complexes

HDAC1 and HDAC2 are the best understood HDAC family members, possessing a myriad of known interactions, yet no comprehensive examination of HDAC1/2 interactions had been performed prior to our study. Our label-free approach identified numerous previously unreported putative interactions (Figure 4, *circles*). As the relative stability of known and unreported interactions has not been systematically examined, we extended our I-DIRT-based relative interaction stability profiling to HDAC1 interactions. In contrast to HDAC3 and class II HDACs, HDAC1 interactions included a mixture of stable and dynamic interactions (Figures 6E and F). To assess whether relative stability measurements are impacted by overall cellular protein abundances, we compared the prey proteome abundance from the PAX database to the relative stability determined by I-DIRT (Supplementary Figure S7). This comparison showed that cellular protein abundance was largely independent of relative I-DIRT stability, suggesting that abundance alone may not be a main contributor to stability. Most known interactions were highly stable and belonged to well-defined complexes, including 10 NuRD and 14 CoREST/ CtBP complex members (Figure 6F, Supplementary Table S5). Among these stable interactions was the recently reported HDAC1 interaction with FAM60A (Munoz *et al*, 2012; Smith *et al*, 2012), which our study confirmed in T cells. A subset of known interactions were less stable and almost exclusively associated with gene transcription, such as the tumor suppressor retinoblastoma-associated protein, RB1, and the DNA-binding zinc-finger protein Ikaros (IKZF1). Interestingly, the only known dynamic interaction that could be ascribed to a specific complex was the histone-lysine *N*-methyltransferase EHMT2, a member of the CtBP co-repressor complex that coordinates deacetylation and H3K9 methylation (Shi *et al*, 2003).

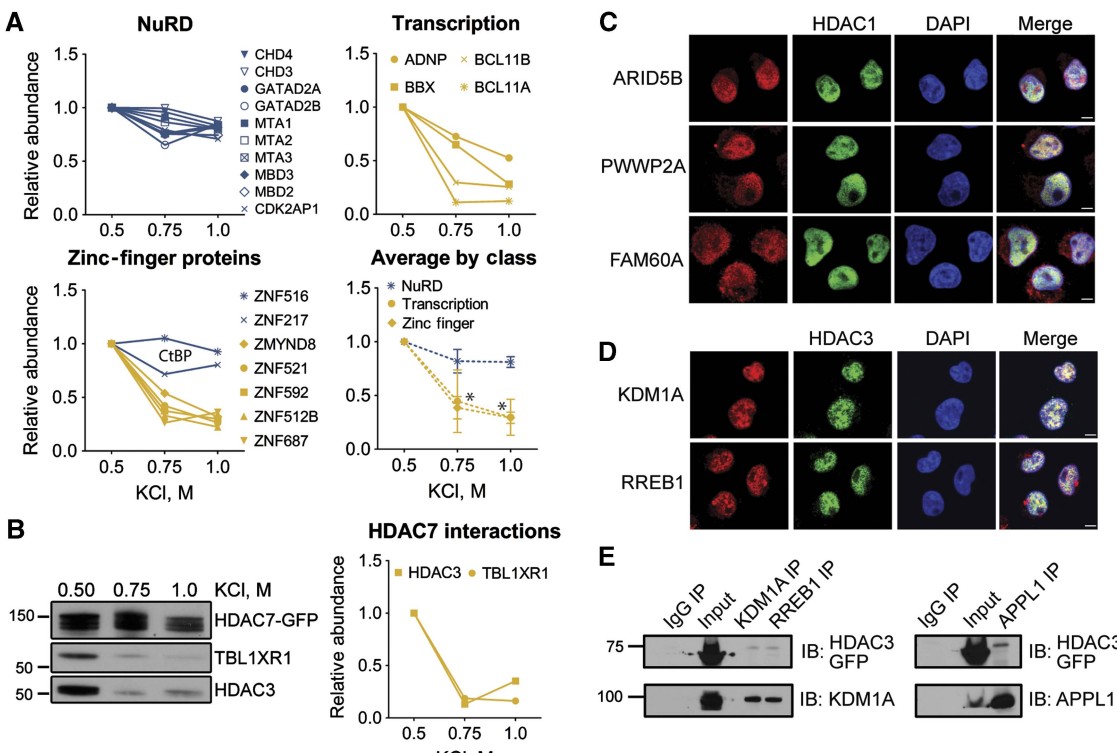

**Figure 7** Biochemical validation and confocal immunofluorescence confirm HDAC interactions identified by proteomics. (**A**) Biochemical validation of HDAC1 interaction stability. The relative abundance of HDAC1 interactions from immunoisolations performed under increasing salt concentration was determined for selected high (blue) and low (yellow) stability proteins comprising the NuRD complex (top left), transcription factors (top right), and zinc-finger proteins (bottom left). The average relative abundance ($\pm$ s.d.) of these classes as a function of [KCl] (*bottom right*) is plotted, excluding ZNF518 and ZNF217 (CtBP complex members). Statistical significance of KCl-dependent relative abundances was assessed compared to the average NuRD-relative abundance (two-way ANOVA, *$P < 0.001$). (**B**) Biochemical validation of HDAC7 interaction stability. The relative abundances of TBL1XR1 and HDAC3 were assessed by western blotting, normalized by densitometry to HDAC7. (**C**) Colocalization of HDAC1–EGFP with ARID5B, PWWP2A, and FAM60A, and (**D**) colocalization HDAC3–EGFP with KDM1A and RREB1 in CEM T-cell lines. Localization of EGFP-tagged HDACs and selected proteins were detected using anti-GFP antibody (green) and antibodies against endogenous proteins (red); DNA is visualized by DAPI (blue); $\times$ 63 oil immersion lens; scale bar 5 μm. (**E**) Reciprocal affinity purifications (IP) for HDAC3–EGFP using antibodies against endogenous KDM1A and RREB1 (*left*), and APPL1 (*right*). EGFP-tagged HDAC3 was detected by western blot. IgG was used as negative control.

In contrast to known interactions, the majority of previously unreported HDAC1 interactions were relatively less stable (Figure 6F). However, similar to known interactions, these fast-exchanging proteins have transcription-related functions, including the zinc-finger protein Helios (IKZF2), HMG box transcription factor BBX, and activity-dependent neuroprotector homeobox protein ADNP. Most putative, stable interactions share similar functional classifications with those that are known and stable. For example, three proteins with methylation-related functions, ARID5B, PWWP2A, and WDR5, were highly stable. These proteins may serve an essential, though as yet unknown, role within HDAC1/2 complexes to coordinate methylation and deacetylation states. Moreover, zinc-finger proteins are prominent members of HDAC1-containing complexes, such as ZNF217 and ZNF215 in the CtBP complex. Our interactome not only provides evidence for additional zinc-finger proteins within HDAC1 complexes, but also demonstrates that these proteins are stable complex members. Interestingly, the one uncharacterized protein identified as highly stable, C16orf87, contains a zinc-binding ribbon domain (PFAM UPF0547). Overall, by integrating metabolic labeling (I-DIRT) with label-free analysis (SAINT), we demonstrate class- and function-dependent differences in relative interaction stabilities of several HDACs. This hybrid approach is not restricted to HDAC interactions but is also suited to profile interaction stability for many different complexes.

To biochemically validate the ability of the I-DIRT/SAINT approach to determine relative interaction stability, we independently isolated HDAC1–EGFP using buffer conditions with increasing KCl concentrations. We assessed the relative abundance of co-isolated interactions by mass spectrometry. As predicted, stable interactions, such as the NuRD complex, remained largely associated with HDAC1 at high-salt concentration (Figure 7A). In contrast, the interactions with transcription factors and zinc-finger proteins were depleted in a dose-dependent manner. A clear validation of our I-DIRT results is shown by the striking difference between the relative stability trends of zinc-finger proteins. The known CtBP complex components, ZNF516 and ZNF217, were stable at high-salt concentrations, while the previously uncharacterized zinc-finger proteins that I-DIRT predicted as less stable interactions (Figures 6E and F) were depleted in a KCl-dependent manner (Figure 7A). Similarly, we performed validation of HDAC7 interaction stability with TBL1XR1 and HDAC3 (Figure 7B). I-DIRT predicted these interactions as less

stable (Figure 6B), in agreement with the nuclear-cytoplasmic shuttling of HDAC7. Upon isolation of HDAC7–EGFP under buffer conditions with increasing KCl concentrations, we observed by western blotting that TBL1XR1 and HDAC3 are diminished in a dose-dependent manner. Altogether, these results support the use of the integrated I-DIRT/SAINT approach for profiling of relative interaction stabilities.

Additionally, as a complement to profiling the interaction stability of specific interactions, this approach can assist in mitigating false-negative interactions. In particular, we observed a small number of interactions across HDAC families that fell below the SAINT specificity threshold, but were classified as stable by I-DIRT (Figures 6A–C and E, *blue triangles*). For class II HDACs, these proteins included the well-known interactions, HDAC3 (in HDAC5 IP) and MEF2C (in HDAC7 IP) (Figures 6A and B). In HDAC1 isolations, BAHD1 (Lebreton *et al*, 2011) and HBP1 (Swanson *et al*, 2004) were among the known interactions, as was a putative interacting partner mortality factor 4-like protein 1 (MORF4L1), a component of the NuA4 histone acetyltransferase complex (Doyon and Cote, 2004) (Figure 6D). These interactions likely did not meet the SAINT specificity threshold due to their low spectral counts, possibly derived from low intracellular abundance and/or poor detectability by mass spectrometry. The identification of well-known interactions within this group of proteins illustrates the utility of I-DIRT for identifying low abundance, yet specific interactions that are not amenable to label-free spectral counting approaches. We also identified a set of proteins common among the four HDACs that had very low SAINT scores ($<0.1$) but maximal I-DIRT ratios ($=1.0$). As this set included abundant, secreted extracellular matrix and cell junction proteins, such as dermicidin, desmoplakin, and desmoglein, it likely reflects extrinsic, environmental contaminants that are only present in the naturally occurring light ($^{12}$C) isotope state.

To further validate the ability of this hybrid approach to identify previously unreported HDAC interactions, we next selected several putative HDAC interactions for further validation with complementary experimental approaches. Focusing on stable interactions, we assessed their cellular localization compared to the associated HDAC. We observed that ARID5B, PWWP2A, and FAM60A co-localized with HDAC1-EGFP in the nucleus of CEM T cells (Figure 7A). We also validated the interaction specificities of ARID5B, FAM60A, and C16orf87 with HDAC1 through reciprocal isolations using antibodies against the endogenous proteins (Supplementary Figure S8). Together, these results indicate that ARID5B, PWWP2A, FAM60A, and C16orf87 are specific and stable interactions of HDAC1.

Moreover, we could also confirm previously unreported HDAC3 interactions that we found to be stable. We observed that both KDM1A and RREB1 co-localize with HDAC3–EGFP in the nucleus of CEM T cells (Figure 7B). Furthermore, reciprocal isolations using antibodies against the endogenous proteins showed that both KDM1A and RREB1 interact with HDAC3–EGFP (Figure 7B). We also observed that KDM1A is co-isolated with immunoaffinity purified endogenous RREB1 (Figure 7C). Taken together with previous reports that RREB1 co-isolates with KDM1A (Shi *et al*, 2005), and that KDM1A co-isolates with HDAC3 (Bantscheff *et al*, 2011), our findings suggest that

KDM1A and RREB1 may represent an HDAC3-containing subcomplex. Interestingly, DCC-interacting protein 13-alpha (APPL1) was identified by SAINT as a specific interacting partner for both HDAC3 (score = 1.0) and HDAC5 (score = 0.98), but was not detected in either I-DIRT experiment. APPL1 has been linked to modulation of HDAC1 deacetylase activity via disruption of HDAC1-NuRD association and recruitment to chromatin regions (Banach-Orlowska *et al*, 2009). As a link to HDAC3 has not been previously reported, we performed reciprocal isolations of endogenous APPL1, validating its association with HDAC3–EGFP (Figure 7C). Therefore, the use of a hybrid label-free and isotope-labeled approach for profiling interaction specificity and stability generated high confidence candidates, leading to a high rate of successful validation by independent experimental approaches. More broadly, application of these holistic approaches to other experimental systems will advance our understanding of protein complex composition, assembly, and regulation by assessing and validating interaction specificity and relative stability.

## Discussion

Though originally identified as histone-modifying enzymes, it is now increasingly clear that the HDAC family of lysine deacetylases (HDAC1–11) possess enzymatic activity towards non-histone substrates. Through the regulation of cellular acetylation states, HDACs are fundamental to numerous cellular processes, including chromatin-remodeling and metabolic-signaling pathways, and can influence human disease progression. Towards understanding HDAC-substrate relationships on a global level, an elegant study was recently published using a genome-wide synthetic lethality screen to identify potential signaling pathways influenced by lysine deacetylases (Lin *et al*, 2012). However, a systematic study of protein–protein interaction profiles across the entire HDAC family had not been conducted. Here, we performed the first global protein interaction network for all 11 human HDACs. This interactome network was assembled from protein–protein interactions in human CEM T cells. Eleven CEM T-cell lines were independently constructed, with stable expression, localization, and activity confirmed for each EGFP-tagged HDAC. To our knowledge, this is the first proteomic study for any histone deacetylase in T cells. A large fraction of the current knowledge of interactions comes from studies of individual HDACs performed in common lab cell lines (e.g., HeLa cells). We selected a CEM T cell line model due to its relevance in immune response, viral infection, and cancers, such as T- and B-cell malignancies, for which the HDAC inhibitor drugs, vorinostat and romidepsin, are currently being employed for treatment. As the molecular mechanisms and mode of action for many HDAC inhibitors are not fully understood, our study provides new molecular targets and HDAC-associated biological functions that can aid in the design of future therapeutic studies. The HDAC1 interactions, B-cell lymphoma/leukemia proteins, BCL11A and BCL11B, are responsible for normal lymphoid development and have a role in lymphoid malignancies (Satterwhite *et al*, 2001; Liu *et al*, 2003). Primarily, it is thought that the BCL11

family has a role in the development of adult T-cell leukemia/lymphoma (ATLL) through their chromosomal amplification and translocation. For example, in an adult T cell leukemia patient, the 5′ region of the BCL11B gene was found fused to intron 3 of the HELIOS gene (Fujimoto *et al*, 2012), which interestingly, we also identified as HDAC1 interaction. While the functional consequences of these protein–HDAC interactions in the development of ATLL remain to be elucidated, our study fills a deficit in knowledge of HDAC interactions in T cells.

## Linking global interaction maps to HDAC-dependent biological processes

An important aspect of our study was the detection of numerous unreported HDAC interactions. Several questions are raised by these data. First, are these interactions part of known (e.g., Sin3a or CoREST) or unknown HDAC-containing complexes? And second, what cellular processes could these specific protein interactions regulate? To address these questions, our study leveraged several complementary experimental approaches: (1) bioinformatics analysis of functional protein networks, (2) proteomic profiling of interaction stability, and (3) targeted siRNA functional assays. Using gene annotation and ontology databases, we assigned biological processes for both known and unreported protein interactions. As would be predicted, many interactions were associated with transcriptional regulation; however, bioinformatics analyses also highlighted cellular processes not traditionally linked to HDACs, such as control of cell cycle, ubiquitination, and RNA processing. One drawback of this approach is incorrect or incomplete functional annotation, which can lead to false positives and biased exclusion of proteins from follow-up studies due to their 'unknown' classification. Continued improvement in the coherent annotation of computational databases and their intelligent integration across various experiment designs will undoubtedly provide significant benefit to future –omics studies.

## Profiling the relative protein interaction stability of HDAC complexes

Given the drawbacks of computational analyses, we designed a complementary proteomic workflow to examine the relative stability of HDAC interactions, which provided a deeper functional understanding of HDAC interactions. For these experiments, we employed two distinct AP-MS analyses, a label-free approach using SAINT scoring and an isotope-labeled approach, I-DIRT. Traditionally, I-DIRT has been used to identify non-specific associations to the isolated protein complexes themselves. We hypothesized that the integration of these two approaches would inform on the relative stability of interactions. Proteins with high SAINT scores but lower I-DIRT ratios would reflect fast exchanging, i.e., dynamic interactions, while proteins with both high-SAINT scores and I-DIRT ratios would be more stable interactions. Indeed, analysis of NCoR complex members, known interactions with class IIa enzymes and HDAC3, demonstrated fast exchange within HDAC5- and HDAC7-containing complexes (ratios ~0.5), yet stable association with HDAC3 (ratios >0.8).

Therefore, this approach is useful for identifying differential relative stability for identical proteins that may exist in distinct complexes. Using this approach for sub-complexes that are shared between different proteins could inform on the degree of connectivity (direct versus indirect) or the subcomplex's distinct functional roles.

We next extended our stability assessments to HDAC1, which had the largest number of SAINT-specific interactions (90 proteins ⩾0.80). One striking result was the finding that members of multiprotein HDAC1/2 complexes were highly stable, likely reflecting the essential nature of the core deacetylase complex (HDAC1/2 and RBBP4/7) within chromatin-remodeling complexes. Given the relatively high stability of these complexes, HDAC1/2 may be concurrently present in these complexes, that is, the core deacetylase complex is largely present as a pre-assembled functional unit, which would allow for more rapid alterations to chromatin structure, and thus more efficient control over cellular fate. However, these highly stable complexes (e.g., NuRD, Sin3) are known to have divergent functions. This raises the question of how HDAC-containing complexes achieve specificity in their chromatin-modifying activities. One possibility is that there are proteins that exist independently of these pre-assembled remodeling complexes that initiate and/or target HDAC-dependent remodeling. It is tempting to speculate that the proteins we identified as fast-exchanging could serve these roles, particularly ones that function as DNA-binding or transcriptional regulators, such as Ikaros (IKZF1) and retinoblastoma-associated protein 1 (RB1). Ikaros is a transcriptional regulator of hematopoietic cell differentiation, which targets NuRD and SWI/SNF complexes to the beta-globulin gene locus in erythrocytes (Kim *et al*, 1999). Notably, our study identified 15 previously unreported interactions that have transcription-related functions, 10 of which were relatively less stable. These proteins included several zinc-finger proteins, the DNA-binding protein Helios, and the HMG box transcription factor BBX, which may play a role in fine-tuning the recruitment of HDAC1 complexes to chromatin.

Since we identified numerous proteins of uncharacterized functions in our interactome study, their relationships to HDAC function(s) are challenging to predict and will certainly require additional study. Among the proteins co-isolated with HDAC1, C16orf87 was of particular interest, resembling a putative zinc-finger domain-containing protein. Several additional HDAC1 interactions were zinc-finger proteins known either to be part of HDAC1/2 complexes or to be involved in chromatin remodeling. A BLAST alignment of C16orf87 showed that its N-terminal domain (amino acids 1–55) shared 93% similarity with the putative zinc-finger-ribbon domain of a predicted SETMAR protein-like isoform 2 from *Canis lupus famliliaris*. SETMAR (Metnase) contains a fusion of the catalytic SET domain, capable of histone methylation (Robertson and Zumpano, 1997). The presence of a zinc-finger domain in C16orf87 suggests that its interaction with HDAC1 might function to target HDAC1 to chromatin via DNA binding activity.

Overall, increasing our understanding of a protein's relative stability within complexes serves to further define its functionality and provides a basis for the temporal relationship between cell signaling events and recruitment of enzymatic

activities (e.g., methylation and deacetylation). Additionally, by using two complementary AP-MS approaches, protein interactions that surpass both scoring thresholds afford a reduced false positive rate of identification, representing both specific and stable interactions. This approach can be readily integrated into AP-MS workflows and applied in a global manner to concurrently examine stability and specificity of a variety of cellular complexes.

## Functional relationships between deacetylation and demethylation

Our HDAC interactome network strengthened a connection between enzymes involved in deacetylation and demethylation processes, best exemplified by the association of KDM1A and RREB1 with HDAC3. KDM1A is a member of the CoREST complex, which couples the deacetylase activity of the HDAC1/HDAC2 dimer core to the demethylase activity of KDM1A (Shi *et al*, 2003, 2005). More recently, KDM1A was reported to associate with the NuRD complex (Wang *et al*, 2009), suggesting a role in several HDAC complexes with varied functions. RREB1 was first identified as a Ras/Raf-responsive transcriptional co-factor with function in cellular differentiation (Zhang *et al*, 1999). Interestingly, RREB1 was co-isolated in a KDM1A purification along with CoREST complex members in HeLa cells (Shi *et al*, 2005). We have shown that both KDM1A and RREB1 interact with HDAC3 and validated these associations using SAINT, I-DIRT, reciprocal isolation, and co-localization. Together, these results suggest that KDM1A, RREB1, and HDAC3 represent a subcomplex whose regulation and transcriptional targets remain to be determined, but that likely employs the deacetylase activity of HDAC3.

Further supporting the theme of coupled deacetylation and demethylation are the newly identified and validated interactions of HDAC1 with the transcription-associated protein ARID5B, and PWWP2A. ARID5B is a DNA-binding transcriptional co-activator that was shown to exist in a complex with the H3K9me2 demethylase PHF2 (Patsialou *et al*, 2005; Baba *et al*, 2011). Similar to PHF2, HDAC1 could associate with ARID5B to facilitate the direct recruitment of this complex to target gene promoters for deacetylation of substrates by HDAC1, in lieu of demethylation by PHF2. Association of ARID5B with these enzymes would therefore serve to coordinate the methylation and acetylation status of the same target promoters via alternate recruitment of PHF2 or HDAC1. The PWWP domain-containing protein, PWWP2A, currently has no known function, yet PWWP domains have been characterized as putative H3K36me3 binding motifs required for transcriptional elongation (Vermeulen *et al*, 2010). Thus, PWWP2A association with HDAC1 could serve to link HDAC1 deacetylation to the H3K36 region. Together, these HDAC1 and HDAC3 associations emphasize the important functional relationship between deacetylation and demethylation.

## HDACs in cell cycle regulation

HDAC activities and functions are thought to be regulated during cell cycle progression, yet the mechanisms involved in cell cycle-dependent regulation of HDACs are not yet understood. Recent studies have shown that class IIa HDACs interactions and phosphorylations are modulated during mitosis. Aurora B kinase phosphorylates these enzymes within their NLS domains, likely as a means to regulate HDAC-mediated transcriptional repression during mitosis (Guise *et al*, 2012). The HDAC interactome provides additional evidence for roles of HDACs in cell cycle regulation. We identify HDAC8 associations with multiple members of the cohesin complex (SMC1A, SMC3, and STAG2). The cohesin complex is responsible for keeping sister chromatids together from the beginning of S phase to the end of anaphase, prior to cell division (Dorsett and Strom, 2012). While these interactions have not been reported for HDAC8, the yeast HDAC8 homolog, Hos1, deacetylates SMC3 following removal of the cohesin complex from sister chromatids (Beckouet *et al*, 2010; Borges *et al*, 2010; Xiong *et al*, 2010). Moreover, a recent study reported that HDAC8 is the vertebrate SMC3 deacetylase (Deardorff *et al*, 2012). Given the mounting evidence for HDAC8 regulation of SMC3 and the cohesin complex, this raises the possibility that other members of the cohesin complex may be regulated by multiple HDACs. Our study supports this hypothesis, identifying association of HDAC11 with SMC1A, SMC3, and STAG1, and HDAC9 with SMC3. Together, our global analysis of HDAC interactions strongly suggests that regulation of the cohesin complex and perhaps other cell cycle events may be influenced by multiple HDAC.

## Involvement of HDAC11 in SMN-dependent splicing

To date, little is known about the protein interactions and functions of HDAC11, the lone class IV HDAC. Here we demonstrate that HDAC11 specifically associates with multiple members of the SMN complex (SMN1, Gemin2, Gemin3, and Gemin4). Interestingly, we observe HDAC11 to be distributed to the perinuclear space in T cells, consistent with its association with SMN complexes. The SMN complex is responsible for spliceosome assembly (Meister *et al*, 2001) and contains SMN1 and several proteins collectively referred to as Gemins (Gemins 2–8) (Fischer *et al*, 1997; Feng *et al*, 2005). SMN1 deficiency in lymphoblasts from patients with spinal muscular atrophy led to splicing defects in U12-type introns from the *ATXN10* gene (Boulisfane *et al*, 2011). We demonstrate that HDAC11 downregulation triggers a similar splicing defect of the U12-type intron (I10) from the *ATXN10* gene. In contrast, no retention events and no difference in splicing efficiency were observed for the *Thoc2* gene, suggesting that HDAC11 has a more subtle effect on intron retention than SMN1 deficiency. HDAC11 may have an indirect role via the SMN complex or a more specialized role in *ATXN10* gene processing. These results establish a previously unreported function for HDAC11 in splicing, suggesting a role in the assembly or stabilization of the SMN complex.

In this study, we present the first comprehensive analysis of protein interactions of the 11 human HDACs, implicating individual HDACs in previously unreported protein complexes and functional pathways. We incorporated proteomics and bioinformatics to systematically and confidently determine interaction specificity. A hybrid approach integrating label-free

and metabolic labeling was designed to determine the relative interaction stability across the HDAC family members. Our results identified previously unreported, stable HDAC interactions and demonstrated that interactions with transcription factors preferentially exist in rapid equilibrium. The resulting interaction networks highlight diverse HDAC functions, including regulation of cell cycle progression. Additionally, we report a previously unknown interaction between HDAC11 and the SMN complex, and demonstrate a functional role for HDAC11 in U12-type mRNA splicing. In summary, global proteomics and targeted functional studies have provided a valuable resource of global HDAC interactions, encompassing the composition and stability of HDAC complexes, insights into the less well-characterized HDACs, and targets for investigating HDAC functions in health and disease states.

# Materials and methods

## Reagents

Antibodies used were an in-house developed rabbit polyclonal anti-GFP (Cristea *et al*, 2005) and a mouse monoclonal anti-GFP (Roche Applied Science). Protein A/G Plus-agarose was purchased from Santa Cruz. Antibodies against SMN1, Dicer1, Gemin3, Gemin4, APPL1, and C16orf87 were purchased from Santa Cruz. Additional antibodies used in this study were anti-HDAC11 (Abgent), anti-HDAC11 (Abcam), anti-KDM1A (Cell Signaling Technology), anti-RREB1 (Bethyl Laboratories Inc.), anti-ARID5B (Bethyl Laboratories Inc.), anti-PWWP2A (Abcam), and anti-FAM60A (Abnova). All other reagents were purchased from Sigma-Aldrich, unless otherwise specified.

## Cloning and construction of EGFP-tagged HDAC cell lines and cell culture

EGFP cDNA (pEGFP-N1; Clontech) was cloned into a pLXSN retroviral vector (Clontech). Flag DNA was then cloned onto the 3′ end of the EGFP, thus generating a pLXSN–EGFP–FLAG plasmid. The plasmids containing HDAC cDNAs were a kind gift of E. Seto (Franco *et al*, 2001; Feng *et al*, 2007). HDAC cDNA was amplified by PCR using HDAC-specific primers (Supplementary Table S6), gel purified, and digested with restriction enzymes. The digestion products were ligated to the 5′ end of EGFP in the above plasmid, thus generating pLXSN–HDAC–EGFP–FLAG retroviral plasmids. All constructs were confirmed by sequencing the coding region using both gene-specific and vector-specific primers. Sequencing of HDAC2 identified three mutations at sites Y167C, T477P, and T480A that we corrected using site-directed mutagenic primers (Supplementary Table S7) (QuikChange Lightning Multi Site-Directed Mutagenesis Kit, Agilent Technologies). To generate CEM T-cell lines stably expressing tagged HDACs, the pLXSN–HDAC–EGFP–FLAG plasmids for each of the 11 HDACs were individually transfected into Phoenix cells using a retrovirus expression system (Orbigen, San Diego, CA) using FuGENE (Roche Applied Science). Transfected Phoenix cells were grown to 90% confluency, and the retrovirus released from the transfected cells into the supernatant was collected by filtration and used to transduce CEM T cells. Cells were selected for stable HDAC expression using G418 (300 µg/ml) (EMD, Gibbstown, NJ) for 2 weeks and sorted by fluorescence-activated cell sorting (Vantage S.E. with TurboSort II; Becton Dickinson, Franklin Lakes, NJ). All CEM T cells were cultured in RPMI-1640 (Gibco) supplemented with 10% fetal bovine serum (Gemini Bio-products) and 1% penicillin/streptomycin (Gemini Bio-products) at 37°C with 5% $CO_2$.

## Confocal microscopy

HDAC–EGFP–expressing CEM T cells were collected by centrifugation at $216 \times g$ for 5 min, washed twice in 2% FBS (v/v)/DPBS (Gibco), and resuspended in 1% BSA (w/v)/DPBS. Resuspended cells were applied to Shandon cytoslides (ThermoFisher) using a Shandon Cytospin 4 Cytocentrifuge (ThermoFisher) operating at $72 \times g$ for 8 min. The cells were fixed with 4% paraformaldehyde at room temperature for 20 min, permeabilized with 0.1% Triton X-100 (v/v) in 0.2% (v/v) Tween-20 in DPBS (PBST) for 15 min, and then blocked in 2% (w/v) bovine serum albumin, PBST at room temperature for 60 min. Incubation with in-house generated anti-GFP was performed at 4°C overnight in blocking buffer. The cells were then washed three times with PBST for 5 min and incubated with a goat–anti-rabbit antibody conjugated to Alexa-488 (Invitrogen). For visualization of the nucleus, the cells were incubated with 1 µg/ml 4',6-diamidino-2-phenylindole (DAPI) in blocking buffer for 15 min, then washed with PBST. Coverslips were then mounted on slides with a drop of Aqua-Poly/Mount media (Polysciences). Confocal images were obtained on a × 63 oil immersion lens on a Leica SP5 confocal microscope. For co-localization experiments, fixed and permeabilized cells were incubated with anti-GFP (in-house generated rabbit polyclonal or mouse anti-GFP) and either rabbit anti-KDM1A, rabbit anti-RREB1, rabbit anti-ARID5B, mouse anti-PWWP2A, mouse, or anti-FAM60A antibodies. Secondary antibodies used were goat-anti-rabbit conjugated to Alexa-488 and goat-anti-mouse conjugated to Alexa568.

## Immunoaffinity purification of HDAC complexes

HDAC1–11 and control EGFP immunoaffinity purifications (IPs) were performed on magnetic beads, as described previously (Cristea *et al*, 2005). CEM T cells stably expressing EGFP alone and HDAC–EGFP were collected by centrifugation at 485*g* for 10 min and then washed once with 25 ml of cold DPBS (Gibco) per 600 ml culture. The washed cell pellet was resuspended in 100 µl/g of cells of 20 mM HEPES-NaOH, pH 7.5, containing 1.2% polyvinylpyrrolidone (w/v) and 1:100 (v/v) protease inhibitor mixture, frozen in liquid nitrogen, and cryogenically lysed using a Retsch MM 301 Mixer Mill ($8 \times 2.5$ min at 30 Hz) (Retsch, Newtown, PA). All further steps were performed at 4°C, unless otherwise stated. The ground cell powder was resuspended in 10 ml/g powder of cold-optimized lysis buffer (Supplementary Table S1), (20 mM HEPES-KOH, pH 7.4, containing 0.11 M KOAc, 2 mM $MgCl_2$, 0.1% Tween-20, 1 µM $ZnCl_2$, 1 µM $CaCl_2$), Triton X-100, NaCl, 10 µg/ml DNase, 1/100 (v/v) protease inhibitor cocktail), followed by the activation of DNase at room temperature for 10 min. The cell suspension was subjected to homogenization using a Polytron ($1 \times 20$ s cycle) (Kinematica), and the cell debris (Pellet) was removed by centrifugation at $8000 \times g$ for 10 min. The supernatant was used for immunoisolation of the HDAC-containing complexes by incubating for 60–75 min with 7 mg of magnetic beads (M270 Epoxy Dynabeads; Invitrogen) that were conjugated with in-house generated rabbit anti-GFP antibody (Cristea *et al*, 2005), The magnetic beads were then washed six times with lysis buffer and twice with DPBS. The washed beads were then incubated with 50 µl of $1 \times$ LDS sample buffer (Invitrogen) for 10 min at 70°C, followed by shaking for 10 min at room temperature. The immunoisolates were recovered and stored at $-20$°C until further processing. Ten percent of the cell pellet, flowthrough and elution (IP) were saved for analysis by western blotting.

For evaluation of relative stability of HDAC1 interactions, immunoisolations were performed with 0.3 g ground cell powder and 4 mg of anti-GFP antibody-conjugated magnetic beads, using lysis buffers containing increasing KCl concentrations (20 mM K-HEPES pH 7.4, 0.11 M KOAc, 0.1% Tween-20 (v/v), 10 µg/ml DNase, 1/100 (v/v) protease inhibitor cocktail, 0.5% Triton X-100 (v/v), containing either 0.5, 0.75, or 1.0 M KCl). Mass spectrometric analysis of the eluates was performed as described below. Similar experiments were performed to validate the stability of HDAC7 interactions with TBL1XR1 and HDAC3, and the presence of isolated proteins at the increasing KCl concentrations were assessed by western blot analyses.

## Western blot analysis

To measure IP efficiency between 1 and 10% of the cell pellet (insoluble fraction after cell lysis), flowthrough (unbound proteins),

and elution (IP) were analyzed by western blotting. Proteins present in the flowthrough were precipitated using 4 volumes of ice-cold acetone overnight. The precipitated proteins were collected by centrifugation at $2000 \times g$ for 10 min, washed once in DPBS, and directly resuspended in 35 µl $1 \times$ SDS sample buffer by vortexing. Samples were subjected to SDS-PAGE electrophoresis and transferred to a PVDF membrane. EGFP-tagged HDACs were detected using an anti-GFP (Roche) antibody.

## Deacetylation activity assays

Deacetylase activity for the HDACs was measured using the Fluor-de-Lys kit (Enzo Life Sciences). Briefly, IPs were carried out using CEM T cells expressing each HDAC alongside a separate GFP control, as described above, except HDAC-containing complexes remained bound to the magnetic bead support. The deacetylase assays were carried out in 50 mM Tris–HCl, pH 8.0, 137 mM NaCl, 2.7 mM KCl, 1 mM $MgCl_2$ and substrate containing an acetylated lysine side chain. The reaction was quenched by addition of developer and the reaction was assayed by fluorescence (excitation 350–380 nm, emission 440–460 nm) using a Synergy Mx fluorometer (Biotek Instruments Inc.). Gen5 microplate data collection software was used for data analysis.

## CEM T-cell culture and affinity purification for I-DIRT analysis

For the isotope-labeled AP-MS workflow, using I-DIRT, wild-type CEM T cells were cultured for at least six population doublings in $^{12}C_6$-Arg- and $^{12}C_6$-Lys-deficient RPMI-1640 that was supplemented with heavy ($^{13}C_6$) Lys and Arg, 10% v/v dialyzed FBS and 1% v/v penicillin/streptomycin. HDAC–EGFP-expressing cell lines were cultured as above, except medium was supplemented with light ($^{12}C_6$) Arg and Lys. Both wild-type and HDAC-expressing cells were frozen as described above. Equal amounts of frozen cells ($\sim$0.5–1.0 g each) were mixed prior to cryogenic lysis. Before incubation with antibody-conjugated beads, 50 µl of the cell lysate was reserved for assessment of the light:heavy mixing ratio in the input. IPs were carried out as described above and then analyzed by AP-MS, as described below.

## Sample preparation and mass spectrometry

HDAC immunoisolates were digested with trypsin in-gel (isotope-labeled workflow) or in-solution (label-free workflow), as previously described with some modification (Guise *et al*, 2012). For in-solution digestion, the filter-aided sample preparation (FASP) method was used (Wisniewski *et al*, 2009). Enzymatic protein digestion was performed in 100 µl of trypsin solution (5 ng/µl in 100 mM ammonium bicarbonate) in Vivacon 500 centrifugal filters (10 kDa MWCO; Sartorius Stedim Biotech, Goettingen, Germany), as described (Kramer *et al*, 2011; Tsai *et al*, 2012). Peptides were were either desalted online using a Magic C18 AQ trap column (3 µm, 100 µm × 2.5 cm, Michrom Bioresources, Inc.) or offline by StageTips (Rappsilber *et al*, 2007) using Empore $C_{18}$ extraction discs (3 M Analytical Biotechnologies). For isotope-labeled I-DIRT experiments, protein samples were reduced with dithiothreitol, alkylated with iodoacetamide, and separated by SDS–PAGE for $\sim$3 cm. Gel lanes containing the partially resolved protein samples were excised, sliced into 1 mm pieces, and combined into 10 total fractions per lane. Proteins were digested with 10 µl of trypsin solution (12.5 ng/µl in 50 mM ammonium bicarbonate) per fraction for 6 h at 37°C, quenched in 0.5% formic acid, and extracted overnight at RT. A second extraction was performed in 50% acetonitrile/0.5% formic acid. Extracted peptides were pooled, concentrated, and desalted as above.

Desalted peptides were analyzed by nanoliquid chromatography–tandem mass spectrometry using a Dionex Ultimate 3000 nRSLC coupled to an LTQ-Orbitrap Velos ETD mass spectrometer (Thermo-Fisher Scientific, San Jose, CA), as previously described (Guise *et al*, 2012). Briefly, peptides from in-gel digests were separated by a 90 min reverse-phase gradient, while in-solution digests were separated by a 180 min reverse-phase gradient. For all analyses, the mass spectrometer was operated in data-dependent acquisition mode. The FT preview scan was disabled and predictive AGC and dynamic exclusion were both enabled (repeat count: 1, exclusion duration: 70 s). A single acquisition cycle consisted of one full-scan mass spectrum ($m/z$ range = 350–1700) in the Orbitrap (30 000 resolution at $m/z$ = 400). CID fragmentation was performed on the top 20 most intense precursor ions with minimum signals of 1E3 in the dual-pressure linear ion trap. Target values for the FT full-scan MS and IT $MS^2$ were 1E6 and 5E3, respectively. All CID fragmentation was performed with an isolation width of 2.0 Th, normalized collision energy of 30, and activation time of 10 ms.

## Protein identification

For the label-free AP-MS workflow, acquired MS data (RAW files) were converted into mzXML open file format files using the Proteowizard conversion tool. mzXML files were searched using the X!Tandem/k-score database search tool against the human subset of the UniProt protein sequence database, appended with a list of common sample contaminants. An equal number of decoy (reversed) sequences were added to the database. X!Tandem searches were performed allowing tryptic peptides only, up to one missed cleavage, 50 p.p.m. monoisotopic precursor ion mass tolerance, and with carboxyamido-methylation of cysteine residues specified as a fixed modification and methionine oxidation as variable modifications. The search results were processed using PeptideProphet and ProteinProphet tools. The data from individual experiments were merged, and the spectral counts were extracted using the software, ABACUS. The combined list of protein identifications (protein groups) was filtered to achieve a protein-level FDR of less than 1%. When computing the spectral counts in individual experiments, proteins were quantified (i.e., spectra counted) if they were identified with a probability equal or greater than 0.9 in that particular replicate.

## SAINT analysis

The spectral count matrix produced by ABACUS was taken for subsequent interaction scoring using SAINT and contained the following information for each prey protein: prey gene name (official gene symbol), protein accession number, protein length, and the spectral counts (total counts) for each purification (or control run). All keratin proteins and external sample contaminants were removed from the data set. The spectral count matrix was reformatted to generate SAINT input files as described in Choi *et al* (2011), and analyzed using SAINT v. 2.3. The following SAINT options were used: lowmode = 0, minford = 1, and norm = 1 (Choi *et al*, 2012). The spectral count of the bait protein in its own purification was set to zero. SAINT was run separately for each HDAC data set, and SAINT results were merged into a single data table using an in-house written script. For each experiment, SAINT computed the individual probability for each biological replicate (iProb). The final SAINT score for each bait-prey pair was then computed as an average of the two highest individual SAINT probabilities (iProb values) for each prey protein in corresponding data set. Identified bait–prey pairs in the HDAC data sets were cross-referenced with previously cataloged HDAC interactions from iRefIndex ver. 10 (Turner *et al*, 2010). ROC-like curves were constructed for HDAC1, 3, and 4. The iRefIndex database was used to plot previously known interactions versus absent interactions as an approximation for true-positive and false-positive rates, respectively. Prey proteins with a SAINT score of $\geqslant$0.90 in HDAC1 and HDAC2, and $\geqslant$0.75 in HDAC3–10 were considered putative protein interactions, while prey proteins with SAINT scores of >0.95 in both HDAC11 biological replicates were considered as putative interactions.

## Protein identification and I-DIRT quantification

For the isotope-labeled AP-MS workflow, tandem mass spectra from acquired MS data (RAW files) were extracted, filtered, and searched as above, except using Proteome Discoverer/SEQUEST (v1.3 Thermo-Fisher Scientific). SEQUEST databases searches were conducted

allowing only tryptic peptides, up to two missed cleavages, 10 p.p.m. monoisotopic precursor ion mass tolerance, 0.5 Da fragment ion mass tolerance, carbamidomethylation of cysteine as a fixed modification, and methionine oxidation, heavy lysine, and heavy arginine as variable modifications. Search results were filtered by *q*-values using Percolator (Kall *et al*, 2007) to achieve a peptide-level FDR of less than 1%. SILAC peptide ratios were calculated by the quantitation module in Proteome Discoverer, normalized by the median SILAC protein ratio of the input sample, and reported as light/ (light + heavy) ratios. Filtered search results were assembled into protein groups, requiring at least two quantified peptides per protein. I-DIRT protein specificity ratios were calculated as the median of individual peptide ratios.

## Hierarchical clustering

SAINT-filtered prey proteins from HDACs 1–10 and their respective $\log_2$-transformed spectral counts were imported into Multiexperiment Viewer software (MeV, ver. 4.8.1). Preys (gene) and baits (HDAC) were clustering by Pearson correlation distance metric (average linkage) using the $\log_2$-transformed spectral counts.

## Construction of interaction and functional networks

Interaction networks were constructed from SAINT-filtered bait–prey pairs and visualized using Cytoscape (Smoot *et al*, 2011). Bait–prey pair relationships that passed the SAINT score thresholds, defined above, were connected by network edges. Ontology and annotation information were downloaded through the Cytoscape interface, and used to group preys into custom biological process categories. The web-based STRING database (Szklarczyk *et al*, 2011) was used to assemble functional networks using a probability score of $\geqslant 0.5$ and default parameters except text mining was disabled. STRING networks were exported to xml format and imported into Cytoscape for network visualization. Enrichment indices for each prey protein were calculated as the ratio between the NSAF and PAX values, as previously described (Tsai *et al*, 2012). For HDAC11 prey proteins that had function associations (Supplementary Table S5 and Supplementary Figure S2), NSAF values were calculated as previously described (Zybailov *et al*, 2007), and PAX values were obtained from http://www.pax-db.org.

## Validation of interactions via reciprocal isolations

Reciprocal IPs were performed using 0.1 g of starting HDAC–EGFP-expressing cell material for each reciprocal isolation experiment. Cells were lysed in buffers equivalent to those used for the original immunopurification of individual HDACs and co-isolating proteins (Supplementary Table S1). Reciprocal immunoisolations were performed as follows. Cell lysates were incubated either with 2 µg of antibody against identified HDAC protein interactions or with 2 µg control IgG for 1 h, followed by a 2 h of incubation with 30 µl of protein A/G Plus-agarose beads (Santa Cruz). Prior to incubation with cell lysates, the agarose beads were prepared by washing twice in lysis buffer, followed by 1 h incubation with lysis buffer containing 2% BSA to reduce non-specific binding. After the immunoisolation, the agarose beads were subsequently washed twice with lysis buffer, twice with DPBS, resuspended in 50 µl of 1 × Laemmli sample buffer, and heated at 95°C for 5 min to elute the proteins from the beads. Eluted protein samples were analyzed by western blotting for both the presence of the target HDAC and isolated protein bait. Immunoisolation of endogenous HDAC11 containing complexes was performed as above, using anti-HDAC11 (Abcam).

## siRNA-mediated knockdown and quantitative real-time PCR

HDAC11 knockdown was performed in WT CEM T cells using siRNA (Sigma-Aldrich), 5′-CGGACAUCACGCUCGCCAU-3′ and 5′-AUGGCGA GCGUGAUGUCCG-3′. Scrambled siRNA was used as a control. Transfection of siRNAs (100 nM) was performed using RNAifect (Qiagen) and cells were harvested 48 h post-treatment. RNA extraction was carried using the EZ RNA extraction kit using manufacturer's instructions (Omega Biotek). Strand cDNA synthesis was performed using reverse transcription protocols detailed in the RETROScript kit (Invitrogen). Real-time quantitative PCR was carried out in a reaction containing cDNA, respective primer pairs (Supplementary Table S8), and SYBR Green PCR Master Mix (Applied Biosystems). GAPDH was used as an internal control for normalization. Primer pairs for U2, U12, U4 and U4atac were designed according to previous studies (Zhang *et al*, 2008). Products from the reaction were separated on a 1.5% agarose gel and visualized using ethidium bromide staining.

## Data availability

The mass spectrometry proteomics data have been deposited to the ProteomeXchange Consortium (http://proteomecentral.proteomexchange.org) via the PRIDE partner repository (Vizcaino *et al*, 2013) with the data set identifier PXD000208, while the protein interactions have been submitted to the IMEx (http://www.imexconsortium.org) consortium through IntAct (Aranda *et al*, 2010) and assigned the identifier IM-18733.

## Supplementary information

## Acknowledgements

We thank lab members that have contributed over the past 5 years to this study with technical assistance and discussions, including HC, DM, TL, JC, YT, AA, RU, L-MB, and AC. We thank JG, CD for technical support (Microscopy and Flow Cytometry Facilities, Princeton University). We also thank the members of the PRIDE team for their assistance in submitting the raw mass spectrometry data and interactions to the ProteomeXchange and Intact repositories. We are grateful for funding from NIH grants DP1DA026192 and R21AI102187, and HFSPO award RGY0079/2009-C to IMC, an NJCCR postdoctoral fellowship to PJ, an NSF graduate fellowship to AJG, and R01-GM-094231 to AIN.

*Author contributions:* PJ, TMG, and IMC designed research; PJ, TMG, AG, YL, FY performed experiments; PJ, TMG, AIN, and IMC analyzed data; PJ, TMG, AG, and IMC wrote the manuscript.

## Conflict of interest

The authors declare that they have no conflict of interest.

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
