## [Review Process File · Molecular Systems Biology]

The functional interactome landscape of the human histone deacetylase family

Preeti Joshi, Todd M. Greco, Amanda J. Guise, Yang Luo, Fang Yu, Alexey L. Nesvizhskii, Ileana M. Cristea

Corresponding author: Ileana M. Cristea, Princeton University

Review timeline:

Submission date:	15 August 2012
Editorial Decision:	22 August 2012
Re-submission:	30 January 2013
Editorial Decision:	07 March 2013
Revision received:	09 April 2013
Accepted:	29 April 2013

Editor: Andrew Hufton/ Maria Polychronidou

Transaction Report:

1st Editorial Decision

22 August 2012

Thank you for having submitted a manuscript entitled "The functional interactome landscape of the human histone deacetylase family" for consideration for publication in Molecular Systems Biology. Your paper has now been seen by Editors of the Journal, and we have decided to return it to you without sending it for extensive peer review.

In this study, you use affinity purification and mass spectrometry to identify binding partners for all eleven human HDAC proteins. Purifications are performed from CEM-T cells expressing tagged HDAC proteins, and the mass spectrometry analysis includes multiple replicates, rigorous statistical analysis of binding specificity, and additional I-DIRT experiments to help identify fast-exchanging interactors. Some of these interactors are confirmed with additional immunoprecipitation and colocalization experiments. These interactions further support connections between HDACs and cell cycle components, and allow you to propose some new functional associations for the different HDACs. We acknowledge that this work provides a detailed dataset of HDAC interactors, and suggests functional differences that may deserve further investigation. At this time, however, we feel that the novel conceptual insights into HDAC protein function remain somewhat modest. Moreover, the mechanistic relevance the most novel interactions for HDAC function, or the ability of these results to advance our understanding of HDACs in human disease, remains somewhat unclear, especially in the absence of more in depth investigation. Overall, we are not convinced that the present study provides the degree of novel biological insight and conclusiveness our audience would expect in Molecular Systems Biology.

I am very sorry to have to disappoint you on this occasion, but I hope that this early decision will allow you to submit your work elsewhere without undue delay.

Re-submission

30 January 2013

January 30, 2013

We are submitting our revised manuscript entitled “**The functional interactome landscape of the human histone deacetylase family**,” for publication as an Article in *Molecular Systems Biology*. Please note that this is a revised manuscript of our original submission (MSB-12-3981). We consider Molecular Systems Biology to be the ideal journal for this significant and large-scale study, and therefore, have opted to perform additional experiments and address all the concerns raised by the editor and resubmit it, rather than submitting it to other journals. In the description below, we highlight the Novelty, Methodology Development, Functional Assays and Insight, and Impact of our study. Sorry for the long message; we just tried to thoroughly address the issues raised by the previous evaluation.

This manuscript is the results of an extensive study, which my lab has been pursuing over the course of five years, presenting the first global protein interactome for all eleven human histone deacetylases (HDAC1-HDAC11). The significance and novelty of our results is further enhanced by the performance of these studies in T cells (see below). Histone deacetylases are essential transcriptional regulators, critically linked to cancer, immune, infectious and cardiac disease. Given their impact on human disease, selected HDACs have been the subject of intense study and are targets for anti-cancer therapy. However, the interactions and functions of many HDACs are not yet fully understood; this knowledge is required for understanding their contribution to different cellular pathways and the future design of therapeutics targeting sub-complexes. The significant findings and novelty of our manuscript are:

GENERAL NOVELTY:

1) We report the first global protein interactome for all eleven human histone deacetylases (HDAC1-HDAC11).

2) This is the first proteomic study for any histone deacetylase in T cells. A large fraction of the current knowledge of interactions comes from studies of individual HDACs performed in common lab cell lines (e.g., HeLa cells). T cell biology critically depends on HDACs for regulating cellular and developmental processes, such as apoptosis, differentiation, and immune response. T cells have important clinical relevance, as small molecules that inhibit HDACs are used for treatment of cutaneous T-cell lymphomas (CTCL). Yet, the presence of off-target effects or development of resistance are key issues. This requires the discovery of more selective targets, such as unique HDAC sub-complexes. To achieve this level of selectivity, we require a better understanding of the ensemble of common and distinct HDAC interactions and their functions as related to T cell biology. Our study fills a significant gap in knowledge of HDAC interactions in T cells.

NOVEL METHODOLOGY:

3) We designed a hybrid approach integrating label-free and isotope-labeling quantification to profile relative interaction stability across co-isolated protein complexes. The ability of this approach to distinguish stable from fast-exchanging HDAC interactions is shown in the **new Figure 6**. We demonstrated class- and function-dependent differences in relative interaction stability of several HDACs. We define previously unreported stable HDAC interactions, and globally demonstrate that well-established chromatin remodeling HDAC1 interactions are largely stable within their complexes,

while transcription factors preferentially exist in rapid equilibrium. This hybrid approach is not restricted to HDAC interactions but would be suited to profile interaction stability for many different complexes.

4) Our study also provides methodological novelty for assessing specificity of protein interactions. We have improved the performance of the label-free-based SAINT algorithm for dealing with heterogeneous datasets with a large dynamic range of protein abundance.

5) By integrating fluorescence microscopy, rapid immunoaffinity purifications (optimized for each HDAC), quantitative mass spectrometry, bioinformatics clustering, functional network analysis, and improved bioinformatics tools for assessing interaction specificity, we identify specific interaction patterns for each HDAC and define common and distinct features of HDAC interactions in T cells. The combination of these orthogonal approaches allowed us to define specific HDAC interactions at high confidence, as demonstrated by the high rate of successful validation and functional relevance of our interactions.

NOVEL FUNCTIONAL ELUCIDATION:

6) To date, HDAC11 remains the least characterized histone deacetylase, with very few reports describing its interactions (<10 interactions in BioGRID) or biological functions. In our study, we demonstrate that HDAC11 is a member of the survival of motor neurons (SMN) complex. The SMN complex interacts with snRNPs and, importantly, contains the disease gene product SMN responsible for the neurodegenerative disorder, spinal muscular atrophy. Next, we performed functional analyses, and showed that HDAC11 down-regulation in T cells triggers a functional U12-type splicing defect, resulting in the accumulation of mis-spliced ATXN10 mRNA. These results do not only provide the first interactome for HDAC11 and its functional association with the SMN complex, but also establish for the first time that HDAC11 is involved in mRNA splicing (**new Figure 5**).

7) HDACs are known to depend on protein interactions to exert their functions. However, the interactions of the majority of these enzymes (HDAC6, 8, 9, 10 and 11) remain poorly characterized. An integrative view of HDAC interactions also lacks for some of the better understood enzymes. Emerging evidence shows that there are still critical protein interactions to be identified. One example is our identification of specific interactions of HDAC1 with FAM60A and HDAC8 with SMC3. Following the first submission of our manuscript to MSB, a high-profile study (Deardorff MA, Nature 2012) showed that knockdown of HDAC8 triggers an increase in SMC3 acetylation, and that loss-of-function HDAC8 mutations impaired cohesin complex regulation, being linked to the congenital malformation disorder, Cornelia de Lange syndrome. Our study demonstrates the association of HDAC8 with SMC3, as well as other members of the cohesion complex in T Cells. Similarly, during the preparation of this manuscript, FAM60A was reported to be a member of the HDAC1-Sin3 complex, responsible for the recruitment of the complex to cyclin D1. These examples further emphasize the significance of our manuscript for discovering novel interactions critical for diverse HDAC functions. In addition to confirming previously reported interactions, our study identified over 200 novel, specific interactions for HDACs 1-11. Interestingly, a subset of these interactions provides important support for the emerging functional relationship between deacetylation and demethylation. We determine for the first time that the lysine demethylase KDM1A and the DNA-binding transcription co-factor

RREB1 exist as part of an HDAC3-containing sub-complex, possibly representing a novel corepressor complex.

Collectively, our work has generated the first global interaction dataset for the eleven human histone deacetylases. The use of global proteomics and targeted functional studies have provided a valuable resource of global HDAC interactions, encompassing the composition and stability of HDAC complexes, insights into less well-characterized HDACs, and targets for investigating HDAC functions in health and disease states. We expect that this study will generate a lot of interest in the scientific community and will be cited accordingly. I am confident that our work fits the high standard and broad readership of *Molecular Systems Biology*.

Thank you for your time and consideration!

Thank you again for submitting your work to Molecular Systems Biology. We have now heard back from the three referees who agreed to evaluate your manuscript. As you will see from the reports below, the referees find the topic of your study of potential interest. They raise, however, substantial concerns on your work, which should be convincingly addressed in a revision of the manuscript.

One of the major points that should be carefully addressed refers to the need to validate the results obtained using the I-DIRT method. Reviewers #1 and #3 include constructive suggestions in this regard.

On a more editorial level, I would kindly ask you to deposit the MS datasets and molecular interaction data in the appropriate public databases. (Additional information is available in the "Guide for Authors" section in our website at <http://www.nature.com/msb/authors/index.html#a3.5.2>) Furthermore, I would like to ask you to include the links and accession numbers in the "Data Availability" section of your manuscript.

If you feel you can satisfactorily deal with these points and those listed by the referees, you may wish to submit a revised version of your manuscript. Please attach a covering letter giving details of the way in which you have handled each of the points raised by the referees. A revised manuscript will be once again subject to review and you probably understand that we can give you no guarantee at this stage that the eventual outcome will be favorable.

REFEREE REPORTS:

Reviewer #1 (Remarks to the Author):

This manuscript describes the human histone deacetylase protein interaction network in T-cells. The authors analyzed the interactions of all 11 human HDACs using state of the art proteomics approaches and the SAINT protein interaction network analysis approach. The authors use imaging and western blotting to validate approaches and particular strengths are the analysis of the poorly characterized HDAC11 protein associations and possibly the interaction stability by combining SAINT and I_Dirt ratios. The body of work will be of interest to a wide research community but the manuscript is in need of major revisions at this time.

The first major concern is the tag itself skewing the results dramatically. The control imaging experiment in Figure 1 appears to demonstrate that EGFP alone is targeted to the nucleus. All of the HDACs appear to be targeted to the nucleus, albeit with different levels of intensity. HDAC4 and HDAC5 appear to have some cytoplasmic localization, for example, but they still are predominantly nuclear. If this tag itself is forcing a skewed localization of all the HDACs to localization the interaction network results would be skewed also and a major effort would be needed to correct for this. All the data in Figure 7 is also largely nuclear. The authors need to explain this carefully in the manuscript since I may be misinterpreting the EGFP results, but the data presented clearly overlaps with DAPI staining. Providing supplemental images where a few proteins with and EGFP tag that should not be in the nucleus do not go into the nucleus would help alleviate this concern.

Another concern is the interpretation and justification of the interaction stability generated from the SAINT score and I-Dirt combination. This is an intriguing possibility and would be valuable if true. Currently it is an interesting observation, but I am not convinced of the argument. The authors should develop in vitro biochemical experiments or perhaps imaging based experiments to show fast and slow exchange of interactions. Validation of this data with a non-proteomic approach would go a long way to supporting this intriguing potential claim.

In the discussion, the manuscript would benefit from framing the presentation of the data with respect to T-cell biology. In reality, this is not a general human HDAC protein interaction network it is the T-cell HDAC protein interaction network. This is a good opportunity for the authors to stress a strength of the study, which is the work was not done in HeLa or HEK293 cells.

One minor concern, but still needs addressing is it is not clear why HDAC11 was excluded from the interaction network. This section needs to be rewritten with a mathematical, for example, justification given for HDAC11 exclusion.

Reviewer #2 (Remarks to the Author):

In this manuscript, Joshi and collaborators report an interaction network including all human HDACs in T-cells. The integrated affinity purification (single GFP immuno-affinity step) and quantitative mass spectrometry to characterize the sets of proteins interacting with 11 human HDACs. The experimental strategy is standard and state-of-the-art (AP/MS). The data analysis is also based on well established strategies (SAINT and integration of STRING data). The experiments are very well controlled and carefully done. This implies an in depth characterization of the baits (EGFP-fusion) in terms of localization and HDAC activity. The authors could demonstrate that the C-terminally EGFP-tagged HDACs are enzymatically active and localize similarly to the wild type versions. The data look solid and imply over 200 previously unreported HDAC interactions. The authors provide further evidence (co-IP and functional assay) for a role of HDAC11 in mRNA splicing. The dataset is likely to be of interest to the scientific community and deserves publication. Several important points though should be addressed:

1) Adaptation of the SAINT algorithm to the HDAC interactome. The cut-off used, i.e. 0.75, 0.9 and 0.95 depending on the HDAC considered (and the numbers of prey found) seems a bit arbitrary. In agreement with this, the authors needed to add back manually MEF2C, a well known HDAC interactors that was filtered out by the procedure. Could it be that too stringent cut-offs have been applied and other real interactors were filtered out? The authors should produce a ROC curve that should easily address the point.

2) Phylogenetic relationships among different HDACs (Figures 3). This part is confusing, as the text does not relate to what is being shown in Figures 3. The authors say "This is consistent with HDAC1 and HDAC2 comprising a catalytic core that functions as part of several multi-protein complexes, including NuRD and CoREST." Is this literature? Then a reference should be provided. Is this results? Then this should be shown in figure 3.

Similarly, "Three class IIa members, HDAC4, 5, and 7, were part of a single cluster, while HDAC9 had a prey protein profile most similar to HDAC8. For this class, clustering was driven by shared interactions with the nuclear co-repressor complex (NCoR) (Figure 3, orange) and the 14-3-3 proteins (Figure 3, purple), which facilitate nucleo-cytoplasmic shuttling of class IIa HDACs (Grozinger & Schreiber, 2000; Kao et al, 2001; McKinsey et al, 2000b; Yang & Gregoire, 2005)." This is completely unclear. First HDAC8 and 9 do not (apparently) co-cluster in Figure 3 and second, neither NCoR nor 14-3-3 can be seen.

More worrisome the authors then, a few sentence later, claim that HDAC6, HDAC8 and HDAC9 are "not part of larger clusters, each forming their own distinct gene cluster (Figure 3, yellow, teal, and blue)". This is apparently contradictory to their statement above.

3) Analysis of the HDAC11 interactome. The figure S2 and the STRING analysis should be clarified, indeed how can they find that NUP153 and Dicer1 interact when none of these proteins have been used as bait. I guess this is not experimental, but STRING data. Then one wonders what the relevance of this interaction/observation here is. Indeed, they (apparently) did not observe the interaction between NUP153 and Dicer, but it was known before. Also in Figure S2 (and 5c) HDAC11 cannot be seen, but these proteins are HDAC11 interactors, this needs clarification.

4) Dynamics of HDAC protein interaction using I-DIRT. It would be interesting to discuss the possible impact of bait-prey stoichiometries on the apparent exchange of differentially labeled prey.

Reviewer #3 (Remarks to the Author):

Review of "The functional interactome landscape of the human histone deacetylase family."

In this paper, the authors identify the cellular proteins that interact with histone deacetylase (HDAC) proteins in human T cells. Interactome datasets were compiled for each of the 11 HDACs via label-free affinity purification followed by MS-MS, and then SAINT analysis. Over 200 novel interactions were identified and HDAC-centric interactomes revealed HDAC involvement in a variety of processes including ubiquitination and cell cycle regulation. Independent analysis of HDAC11 revealed association with RNA editing and processing, specifically with the SMN subnetwork. Downregulation of HDAC11 was associated with accumulation of splicing defects in ATXN10 gene. The authors complemented their finding with a metabolic labeling approach, I-

DIRT, to assess the relative stability of interactions and the binding characteristics of each of the HDACs.

This paper is very impactful in that it provides the first comprehensive and global interactome for HDAC proteins. We agree with the authors that this is particularly important given that HDACs are the targets of several drugs in clinical trials, but the ability to assess their global effect is impaired due to lack of knowledge about the biological pathways and systems in which HDACs are involved. Additionally, the authors have developed a hybrid proteomic approach that can be adapted to the study of other protein interaction networks. By using numerous examples from the literature, they show that the complementarity of the two approaches raises the confidence level of the interactions identified. Overall the paper is well researched, contains a high degree of novelty and is a useful resource platform for further investigations into the HDAC protein family.

Major comments

The authors should find a way to validate the I-DIRT method. They show binding by two proteins by IP and by colocalization, but they do not show a way to validate their binding strength. This can be done, for example, by salt extraction.

In the HDAC11 analysis only one intron of the ATXN10 gene is shown to be misspliced while the wording of the relevant section claims that SMN1 deletion perturbs the splicing of more than one U12 intron (Boulisfane et al, 2011). Did the authors check any of the other introns? If there wasn't intron retention, was there a difference in splicing efficiency? Can the authors check some other genes with U12 introns to show that this is a general phenomenon?

Minor comments:

In section "Protein interaction clustering reflect phylogenetic relationships and functional commonalities among distinct HDACs": The authors write "HDAC bait spectral counts were removed to prevent clustering bias", but the heatmap figure contains HDAC proteins and spectral counts even in the vector representing the experiment in which they should be the bait (e.g. HDAC1 has a yellow color in columns one and two representing the HDAC 1-1 and HDAC1-2 affinity assays). This should be clarified.

- Can the authors explain why the interactome and heatmap don't contain histone proteins? The interaction table in the Excel file contains histone proteins for several of the HDACs but they are not shown in the heatmap.

- "Clustering of biological replicates showed highest similarity to each other" - this is not so for HDAC5-1, HDAC5-2, or HDAC4-3. This sentence should be less absolute. Also, why do some HDACs have three biorepeats? Should be clarified.

In wording referring to Figure 6, the gene HBP1 (Swanson et al, 2004) is mentioned, but it is not in the figure.

The reference to Figure S4 should be S3.

In Figure 5B, the meaning of the clump of green circles at the bottom is unclear.

In Figure 6C,D the Y axis is named differently. In C it's "SAINT score" and in D it's "SAINT probability". For D, the graph is unnecessary since it contains no points. It's enough to have the color bar with a number legend for the I-DIRT stability ratio and then the chart on the right of the HDACs vs. other proteins.

Figure 2A - where does the metabolic -labeled affinity purification come in to play? The box says LC-MS/MS Analysis...

April 9, 2013

We appreciate the valuable insight and positive comments of the reviewers, and have revised our manuscript to include all suggested changes. We have performed all suggested experiments and have included 6 new figures, one as a new Figure 7 and 5 new supplementary figures (Figs. S1, S3, S4, S6, and S7). Importantly, as requested by you and the reviewers, we have provided experimental validation of the I-DIRT results, demonstrating that this is a new approach for assessing relative protein interaction stability. Overall, we have addressed all raised concerns, and as a result the manuscript is significantly improved and suitable for publication in *Molecular Systems Biology* journal. We provide the point-by-point description of the changes we have made in response to the reviewer's specific suggestions at the end of this file. Additionally, as requested, the mass spectrometry proteomics data and molecular interactions have been deposited to ProteomeXchange Consortium (PXD000208) and IntAct (IM-18733), respectively.

As a reminder of the topic and main findings that our manuscript presents, this manuscript is the result of an extensive study, presenting the first global protein interactome for all eleven human histone deacetylases (HDAC1-HDAC11). As recognized by the reviewers, the significance of our results is further enhanced by the performance of these studies in human T cells. Histone deacetylases are essential transcriptional regulators, critically linked to cancer, immune, infectious and cardiac disease. Given their impact on human disease, selected HDACs have been the subject of intense study and are targets for anti-cancer therapy. However, the interactions and functions of many HDACs are not yet fully understood; this knowledge is required for understanding their contribution to different cellular pathways and the future design of therapeutics targeting sub-complexes. The significant findings and novelty of our manuscript are:

GENERAL NOVELTY:

- 1) We report the first global protein interactome for all eleven human histone deacetylases (HDAC1-HDAC11).
- 2) This is the first proteomic study for any histone deacetylase in T cells. A large fraction of the current knowledge of interactions comes from studies of individual HDACs performed in common lab cell lines (e.g., HeLa cells). T cells have important clinical relevance, as small molecules that inhibit HDACs are used for treatment of cutaneous T-cell lymphomas (CTCL). Yet, the presence of off-target effects or resistance are key issues. This requires the discovery of more selective targets, such as unique HDAC sub-complexes. Our study fills a significant gap in knowledge of HDAC interactions in T cells.

NOVEL METHODOLOGY:

- 3) We designed a hybrid approach integrating label-free and isotope-labeling quantification to profile relative interaction stability across co-isolated protein complexes. This approach is not restricted to HDAC interactions, and would be suited to profile interaction stability for many different complexes.
- 4) Our study also provides methodological novelty for assessing specificity of protein interactions. We have improved the performance of the label-free-based SAINT algorithm for dealing with heterogeneous datasets with a large dynamic range of protein abundance.
- 5) By integrating fluorescence microscopy, immunoaffinity purifications, quantitative mass spectrometry, bioinformatics clustering, functional network analysis, biochemistry approaches, and improved bioinformatics tools, we identified specific interaction patterns for each HDAC and defined common and distinct features of HDAC interactions in T cells.

NOVEL FUNCTIONAL ELUCIDATION:

- 6) To date, HDAC11 remains the least characterized histone deacetylase, with very few reports describing its interactions (<10 interactions in BioGRID) or biological functions. In our study, we demonstrate that HDAC11 is a member of the survival of motor neurons (SMN) complex. The SMN complex interacts with snRNPs and, importantly, contains the disease gene product SMN responsible for the neurodegenerative disorder, spinal muscular atrophy. We performed functional analyses and showed that HDAC11 down-regulation in T cells triggers a functional U12-type splicing defect, resulting in the accumulation of mis-spliced ATXN10 mRNA. These results provide the first interactome for HDAC11, revealing a functional association with the SMN complex and establishing for the first time that HDAC11 is involved in mRNA splicing.

7) HDACs are known to depend on protein interactions to exert their functions. However, the interactions of the majority of these enzymes (HDAC6, 8, 9, 10 and 11) remain poorly characterized. An integrative view of HDAC interactions also lacks for some of the better understood enzymes. Emerging evidence shows that there are still critical protein interactions to be identified. One example is our identification of specific HDAC8-SMC3 interaction. Following the first submission of our manuscript to MSB, a high-profile study (Deardorff MA, Nature 2012) showed that knockdown of HDAC8 triggers an increase in SMC3 acetylation, and that loss-of-function HDAC8 mutations impaired cohesin complex regulation, being linked to the congenital malformation disorder, Cornelia de Lange syndrome. Our study demonstrates the association of HDAC8 with SMC3 and other members of the cohesion complex in T Cells. This example further emphasize the significance of our manuscript for discovering novel interactions critical for diverse HDAC functions. In addition to confirming previously reported interactions, our study identified over 200 novel, specific interactions for HDACs 1-11. Interestingly, a subset of these interactions provides important support for the emerging functional relationship between deacetylation and demethylation.

Collectively, our work has generated the first global interaction dataset for the eleven human histone deacetylases. The use of global proteomics and targeted functional studies have provided a valuable resource of global HDAC interactions, encompassing the composition and stability of HDAC complexes, insights into less well-characterized HDACs, and targets for investigating HDAC functions in health and disease states. We expect that this study will generate a lot of interest in the scientific community and will be cited accordingly. I am confident that our work fits the high standard and broad readership of *Molecular Systems Biology*.

Thank you for your time and consideration!

Sincerely,
Ileana Cristea (Princeton U.) & Alexey Nesvizhskii (U. Michigan)

Point-by-point response to reviewers:

Thank you for your careful consideration of our manuscript number MSB-13-4361, entitled “The functional interactome landscape of the human histone deacetylase family”. We appreciate the valuable insight and positive comments of the reviewers, and have revised our manuscript to include all suggested changes. We have performed all suggested experiments and have included 6 new figures, one as a new Figure 7 and 5 new supplementary figures (Figs. S1, S3, S4, S6, and S7). Overall, we have addressed all raised concerns, and as a result the manuscript is significantly improved and suitable for publication in *Molecular Systems Biology* journal. Below is provided a point-by-point description of the changes we have made in response to the reviewer’s specific suggestions. Page numbers in our responses refer to the revised manuscript; reviewer’s comments are marked with “>” and our responses are shown in italics. Thank you again for your time.

Reviewer #1 (Remarks to the Author):

This manuscript describes the human histone deacetylase protein interaction network in T-cells. The authors analyzed the interactions of all 11 human HDACs using state of the art proteomics approaches and the SAINT protein interaction network analysis approach. The authors use imaging and western blotting to validate approaches and particular strengths are the analysis of the poorly characterized HDAC11 protein associations and possibly the interaction stability by combining SAINT and I_Dirt ratios. The body of work will be of interest to a wide research community but the manuscript is in need of major revisions at this time.

>The first major concern is the tag itself skewing the results dramatically. The control imaging experiment in Figure 1 appears to demonstrate that EGFP alone is targeted to the nucleus. All of the HDACs appear to be targeted to the nucleus, albeit with different levels of intensity. HDAC4 and HDAC5 appear to have some cytoplasmic localization, for example, but they still are predominantly nuclear. If this tag itself is forcing a skewed localization of all the HDACs to localization the interaction network results would be skewed also and a major effort would be needed to correct for this. All the data in Figure 7 is also largely nuclear. The authors need to explain this carefully in the manuscript since I may be misinterpreting the EGFP results, but the data presented clearly overlaps with DAPI staining. Providing supplemental images where a few proteins with and EGFP tag that should not be in the nucleus do not go into the nucleus would help alleviate this concern.

The localization of a tagged protein is an important aspect on which we have focused significant attention and have carefully assessed in our initial studies of HDACs. We have experience with functionally assessing protein tagging, and have established that the EGFP tag does not lead to an increased nuclear localization. We present several lines of evidence for this within our HDAC study (shown in Figure 1 and new Figure S1): 1) the sole cytoplasmic localization of HDAC6-EGFP, 2) the dual nuclear and cytoplasmic localizations of class IIa HDACs, 3) dual localization of HDAC10 and HDAC11, and 4) the pan-cellular localization of the EGFP control cell line. To clarify these observations, as suggested by the reviewer, we have included supplementary figures that clearly demonstrate the cytoplasmic and pan-cellular localizations mentioned above (new Fig. S1). Additionally, it is noteworthy to emphasize that these studies are performed in T cells. CEM T-cells are suspension cells, in which the nucleus occupies the majority of the cell volume, and the cytoplasm is present as a relatively thin layer around the nucleus.

We have added this new information and discussion on pages 7-8:

“To confirm that the EGFP tag does not interfere with subcellular localization, we examined each tagged HDAC by immunofluorescence microscopy. Expression of the EGFP tag alone demonstrates that EGFP displays a diffuse localization to both the nucleus and the cytoplasm in CEM T-cells (Figure 1 and S1A), similar to our previous observations in EGFP HEK293 cell lines (Figure S1A). Class I HDACs have been reported to localize to the nucleus, which was mimicked in our observations of EGFP-tagged HDAC localizations. Class IIa enzymes are known to shuttle between the nucleus and the cytoplasm, allowing these HDACs to interact with both nuclear and cytoplasmic proteins in a localization-dependent manner (Greco et al, 2011; Grozinger & Schreiber, 2000; McKinsey et al, 2000a; Paroni et al, 2007; Zhao et al, 2001). Consistent with their known shuttling ability, the EGFP-tagged HDAC4, 5, 7, and 9 were distributed to both nuclear and cytoplasmic compartments, with HDAC4 showing increased cytoplasmic localization when compared to the remaining class IIa HDACs, which were instead predominately nuclear and only partially cytoplasmic. A similar dual localization phenotype was observed for the class IIb enzyme HDAC10, while HDAC6 was predominately cytoplasmic (Figure 1), consistent with previous reports of HDAC6 as a cytoplasmic deacetylase (Hubbert et al, 2002). While the morphology of CEM T-cells grown in suspension provides a minor challenge in visualizing cytoplasmic proteins, as the nucleus occupies a substantial fraction of the total cell volume, close examination illustrates the HDAC6 cytoplasmic enrichment and the dual localizations detailed above (Figure S1B). Therefore, while HDAC localizations have not previously been fully characterized in T-cells, our results agree with endogenous protein localizations reported for HDACs in various cell types (Keedy et al, 2009; Yang & Seto, 2008).”

>Another concern is the interpretation and justification of the interaction stability generated from the SAINT score and I-Dirt combination. This is an intriguing possibility and would be valuable if true. Currently it is an interesting observation, but I am not convinced of the argument. The authors should develop in vitro biochemical experiments or perhaps imaging based experiments to show fast and slow exchange of interactions. Validation of this data with a non-proteomic approach would go a long way to supporting this intriguing potential claim.

We agree with the reviewer that the use of the integrated I-DIRT/SAINT approach for determining relative interaction stability is very valuable and can be utilized when studying diverse complexes, and that these findings could be strengthened by additional experiments. To validate our I-DIRT observations, we have designed and successfully performed two experiments:

- 1. We performed a biochemical validation of HDAC1 interaction stability. We isolated HDAC1-EGFP using lysis buffer conditions with increasing salt concentrations. To determine the relative resistance of the interactions to the increased salt concentrations, we analyzed the co-isolated proteins by mass spectrometry. Stable interactions, such as the NuRD complex, remained largely associated with HDAC1 at high salt concentration. In contrast, the interactions with transcription factors and zinc finger proteins were depleted in a dose-dependent manner. A clear validation of our I-DIRT results is given by the striking difference between the trends of the zinc finger proteins. The known CtBP complex components, ZNF516 and ZNF217, were stable at high salt concentrations, while the previously uncharacterized zinc finger proteins that I-DIRT predicted as less stable interactions were depleted in a KCl-dependent manner. We illustrate these results in the new Figure 7A.***
- 2. Similarly, we performed additional validation of HDAC7 interaction stability with TBL1XR1 and HDAC3. I-DIRT predicted these interactions as less stable, in agreement with the nuclear-cytoplasmic shuttling of HDAC7. Upon isolation of HDAC7-EGFP under lysis buffer conditions with increasing KCl concentrations, we demonstrate by Western blotting that these HDAC7 interactions are diminished in a dose-dependent manner. We illustrate these results in the new Figure 7B.***

We have included these new Results on pg 21-22:

“To biochemically validate the ability of the I-DIRT/SAINT approach to determine relative interaction stability, we independently isolated HDAC1-EGFP using lysis buffer conditions with increasing KCl concentrations. We assessed the relative abundance of co-isolated interactions by mass spectrometry. As predicted, stable interactions, such as the NuRD complex, remained largely associated with HDAC1 at high salt concentration (Figure 7A). In contrast, the interactions with transcription factors and zinc finger proteins were depleted in a dose-dependent manner. A clear validation of our I-DIRT results is shown by the striking difference between the relative stability trends of zinc finger proteins. The known CtBP complex components, ZNF516 and ZNF217, were stable at high salt concentrations, while the previously uncharacterized zinc finger proteins that I-DIRT predicted as less stable interactions (Figure 6E-F) were depleted in a KCl-dependent manner (Figure 7A). Similarly, we performed validation of HDAC7 interaction stability with TBL1XR1 and HDAC3 (Figure 7B). I-DIRT predicted these interactions as less stable (Figure 6B), in agreement with the nuclear-cytoplasmic shuttling of HDAC7. Upon isolation of HDAC7-EGFP under lysis buffer conditions with increasing KCl concentrations, we observed by Western blotting that TBL1XR1 and HDAC3 are diminished in a dose-dependent manner. Altogether, these results support the use of the integrated I-DIRT/SAINT approach for profiling relative protein interaction stabilities.”

>In the discussion, the manuscript would benefit from framing the presentation of the data with respect to T-cell biology. In reality, this is not a general human HDAC protein interaction network it is the T-cell HDAC protein interaction network. This is a good opportunity for the authors to stress a strength of the study, which is the work was not done in HeLa or HEK293 cells.

We thank the reviewer for pointing this out, as we believe that carrying out these experiments in T cells brings an important additional significance to the study. As the reviewer suggested, we have now added discussion to emphasize this aspect (pg. 24-25).

“This interactome network was assembled from protein-protein interactions in human CEM T-cells. Eleven CEM T-cell lines were independently constructed, with stable expression, localization, and activity confirmed for each EGFP-tagged HDAC. To our knowledge, this is the first proteomic study for any histone deacetylase in T-cells. A large fraction of the current knowledge of interactions comes from studies of individual HDACs performed in common lab cell lines (e.g., HeLa cells). We selected a CEM T-cell line model due to its relevance in immune response, viral infection, and cancers, such as T- and B-cell malignancies, for which the HDAC inhibitor drugs, vorinostat and romidepsin, are currently being employed for treatment. Since the molecular mechanisms and mode of action for many HDAC inhibitors are not fully understood, our study provides new molecular targets and HDAC-associated biological functions that can aid in the design of future therapeutic studies. The HDAC1 interactions, B-cell lymphoma/leukemia proteins, BCL11A and BCL11B, are responsible for normal lymphoid development and play a role in lymphoid malignancies (Liu et al, 2003; Satterwhite et al, 2001). Primarily, it is thought that the BCL11 family plays a role in the development of adult T-cell leukemia/lymphoma (ATLL) through their chromosomal amplification and translocation. For example, in an adult T-cell leukemia patient the 5' region of the BCL11B gene was found fused to intron 3 of the HELIOS gene (Fujimoto et al, 2012), which interestingly, we also identified as HDAC1 interaction. While the functional consequences of these protein-HDAC interactions in the development of ATLL remain to be elucidated, our study fills a deficit in knowledge of HDAC interactions in T cells.”

>One minor concern, but still needs addressing is it is not clear why HDAC11 was excluded from the interaction network. This section needs to be rewritten with a mathematical, for example, justification given for HDAC11 exclusion.

We analyzed the HDAC11 interactome separately due to (1) the lack of knowledge regarding HDAC11 biology and interactions (less than a dozen citations in Pubmed) and (2) the relatively large number of identified putative HDAC11 interactions size. We agree that this aspect was not adequately depicted in the original manuscript. To better integrate HDAC11 within the interactome of the other HDACs, we now have included a Supplementary Figure S4 to show the common interactions (23 out of 124) between HDAC11 and the interactions from the other HDAC1-10.

This point was included in the Results on pg 16:

“The enriched functional attributes of HDAC11 interactions are consistent with the limited overlap of the putative protein interactions with the other HDACs (Figure S4).”

Reviewer #2 (Remarks to the Author):

>In this manuscript, Joshi and collaborators report an interaction network including all human HDACs in T-cells. The integrated affinity purification (single GFP immuno-affinity step) and quantitative mass spectrometry to characterize the sets of proteins interacting with 11 human HDACs. The experimental strategy is standard and state-of-the-art (AP/MS). The data analysis is also based on well established strategies (SAINT and integration of STRING data). The experiments are very well controlled and carefully done. This implies an in depth characterization of the baits (EGFP-fusion) in terms of localization and HDAC activity. The authors could demonstrate that the C-terminally EGFP-tagged HDACs are enzymatically active and localize similarly to the wild type versions. The data look solid and imply over 200 previously unreported HDAC interactions. The authors provide further evidence (co-IP and functional assay) for a role of HDAC11 in mRNA splicing. The dataset is likely to be of interest to the scientific community and deserves publication.

We appreciate the reviewer’s positive comments and acknowledgment of its significant interest to the scientific community.

Several important points though should be addressed:

>1) Adaptation of the SAINT algorithm to the HDAC interactome. The cut-off used, i.e. 0.75, 0.9 and 0.95 depending on the HDAC considered (and the numbers of prey found) seems a bit arbitrary. In agreement with this, the authors needed to add back manually MEF2C, a well known HDAC interactors that was filtered out by the procedure. Could it be that too stringent cut-offs have been applied and other real interactors were filtered out? The authors should produce a ROC curve that should easily address the point.

The SAINT cut-offs were selected to adequately balance sensitivity and specificity. We used the iRefIndex database of literature curated interactions to guide our selection of SAINT score thresholds. We now provide as a new supplemental figure S3, “ROC” curves for representative HDACs (HDAC1, 3, and 4) that had the largest number of literature curated interactions. Since (1) database protein interactions are curated across different model system and (2) not all HDAC interactions are known (exemplified by our analysis of HDAC1), true positive rates may be underestimated, while false positive rates may be overestimated. Therefore, these models should be interpreted cautiously. Moreover, since we extended SAINT scoring filters to HDAC interactions that have not been well characterized, we selected initial SAINT score thresholds that were more conservative, and were only relaxed upon integration with the I-DIRT AP-MS approach. Yet it is important to note that even at these relatively stringent thresholds, we identified between 40 – 60% of known interactions in the iRefIndex (Figure S3A-C), which includes interactions identified in various cell types and experimental conditions.

We have added the following sentences to the Methods (pg 39) to explain this analysis:

“Identified bait-prey pairs in the HDAC datasets were cross-referenced with previously cataloged HDAC interactions from iRefIndex ver. 10 (Turner et al, 2010). ROC-like curves were constructed for HDAC1, 3, and 4. The iRefIndex database was used to plot previously known interactions vs. absent interactions as an approximation for true positive and false positive rates, respectively.”

We also include an explanation of this selection of SAINT thresholds in the Results (pg 11):

“Selection of SAINT scoring thresholds were aided by generating ROC-like curves for HDAC1, 3, and 4, for which the greatest number of HDAC interactions have been catalogued. We determined putative protein interactions at different SAINT scoring thresholds. Then, by comparison to the iRefIndex database (Turner et al, 2010), we approximated true positive and false positive rates based on presence or absence in the iRefIndex database, respectively (see Figure S3). Since we do not have reliable estimates of error rates for the lesser studied HDACs, our selection of initial SAINT scoring thresholds were conservative. We considered prey proteins with an average score of ≥ 0.75 in at least one HDAC isolation as putative specific interactions, except for HDAC1/2 and HDAC11 preys, which required an increased stringency of ≥ 0.90 (see Figure S3A) and > 0.95 , respectively. Although MEF2C, a well-known interaction among the class II HDACs (Lu et al, 2000), was identified in the HDAC immunisolates, given its lower abundance it

did not generate significant spectral counts to pass the SAINT score filters (**Table S2**) and was therefore manually included in subsequent analyses. Yet, even at these relatively stringent thresholds, we identified between 40 – 60% of known interactions in the iRefIndex (**Figure S3A-C**), which includes interactions identified various cell types and experimental conditions.”

>2) Phylogenetic relationships among different HDACs (Figures 3). This part is confusing, as the text does not relate to what is being shown in Figures 3. The authors say "This is consistent with HDAC1 and HDAC2 comprising a catalytic core that functions as part of several multi-protein complexes, including NuRD and CoREST." Is this literature? Then a reference should be provided. Is this results? Then this should be shown in figure 3.

The presence of HDAC1 and HDAC2 as a core for several functional complexes has been well established and documented by numerous studies.

As requested, we have included the literature references that support this statement on pg 12:

“This is consistent with HDAC1 and HDAC2 comprising a catalytic core that functions as part of several multi-protein complexes, including NuRD (Tong et al, 1998) and CoREST (You et al, 2001).”

>Similarly, "Three class IIa members, HDAC4, 5, and 7, were part of a single cluster, while HDAC9 had a prey protein profile most similar to HDAC8. For this class, clustering was driven by shared interactions with the nuclear co-repressor complex (NCoR) (Figure 3, orange) and the 14-3-3 proteins (Figure 3, purple), which facilitate nucleo-cytoplasmic shuttling of class IIa HDACs (Grozingler & Schreiber, 2000; Kao et al, 2001; McKinsey et al, 2000b; Yang & Gregoire, 2005)." This is completely unclear. First HDAC8 and 9 do not (apparently) co-cluster in Figure 3 and second, neither NCoR nor 14-3-3 can be seen.

We thank the reviewer for noting the inaccuracy of the second half of the first sentence. Indeed, Figure 3 does not clearly support a significant commonality between HDAC9 and HDAC8 interactions.

The statement on pg 13 has been modified accordingly: "Three class IIa members, HDAC4, 5, and 7, were part of a single cluster. For this class, clustering was driven by shared interactions with the nuclear co-repressor complex (NCoR)..."

>More worrisome the authors then, a few sentence later, claim that HDAC6, HDAC8 and HDAC9 are "not part of larger clusters, each forming their own distinct gene cluster (Figure 3, yellow, teal, and blue)". This is apparently contradictory to their statement above.

Now that the previous comment has been addressed, our assessment of HDAC6, 8, and 9 in Figure 3 is correct and consistent with previous statements.

>3) Analysis of the HDAC11 interactome. The figure S2 and the STRING analysis should be clarified, indeed how can they find that NUP153 and Dicer1 interact when none of these proteins have been used as bait. I guess this is not experimental, but STRING data. Then one wonders what the relevance of this interaction/observation here is. Indeed, they (apparently) did not observe the interaction between NUP153 and Dicer, but it was known before. Also in Figure S2 (and 5c) HDAC11 cannot be seen, but these proteins are HDAC11 interactors, this needs clarification.

As the reviewer noted, Figure S2 and Fig 5C are functional protein networks assembled by STRING analysis. While the network nodes were derived from the HDAC11 candidate interactions, the edges are functional associations retrieved from the STRING database.

Towards relevance of this analysis, we chose to perform STRING analysis and overlay experimental enrichment indices for several purposes. First, from our previous publication on the interactions of another deacetylase, SIRT7, (see ref Tsai et al, MCP; PMID: 22147730) our lab has demonstrated its usefulness in identifying proteins that may exist within common/related complexes. Second, HDAC11 is arguably the least well studied HDAC (only ~6 publications directly to its name, and 9 interactions in the BIOGRID database). Therefore, to gain insight into its potential cellular functions, we examined known functional relationships between its candidate interactions.

Since HDAC11 was the only bait protein in the context of our proteomic analysis of HDAC11 interactions, this experiment cannot provide direct experimental evidence for any binary relationship, including NUP153-Dicer. For this reason, HDAC11 was not integrated into the STRING network as this would suggest a misleading relationship. However, this was not our goal for this analysis. Rather, we wanted to identify high value targets for follow-up

studies. This is supported by our ultimate finding with targeted experiments that HDAC11 interacts with an SMN complex and has a functional effect on mRNA splicing. To better integrate HDAC11 within the interactome of the other HDACs, we have now provided a new supplementary figure (Fig. S4) that shows the connectivity of HDAC11 interactions.

>4) Dynamics of HDAC protein interaction using I-DIRT. It would be interesting to discuss the possible impact of bait-prey stoichiometries on the apparent exchange of differentially labeled prey.

We agree that the stoichiometry of proteins with HDAC complexes is an important aspect of their regulation. We cannot fully address this issue in our current experimental approach that uses relative quantification techniques (i.e., I-DIRT/SILAC). However, we did further investigate a similar idea, i.e., the relationship between cellular abundance and stability. As shown in new supplementary Figure S7, for HDAC1, we compared the prey protein abundances at the proteome level using the PAX database to the relative stability determined by I-DIRT. This comparison showed that cellular protein abundance was largely independent of relative I-DIRT stability, suggesting that abundance alone may not be a main contributor to stability. However, more detailed studies of stoichiometry using alternative quantification techniques would be required to understand this relationship. We feel this is beyond the scope of the current study.

We have added description of this result on pg. 20.

"To assess whether relative stability measurements are impacted by overall cellular protein abundances, we compared the prey proteome abundance from the PAX database to the relative stability determined by I-DIRT (Figure S7). This comparison showed that cellular protein abundance was largely independent of relative I-DIRT stability, suggesting that abundance alone may not be a main contributor to stability."

Reviewer #3 (Remarks to the Author):

Review of "The functional interactome landscape of the human histone deacetylase family."

In this paper, the authors identify the cellular proteins that interact with histone deacetylase (HDAC) proteins in human T cells. Interactome datasets were compiled for each of the 11 HDACs via label-free affinity purification followed by MS-MS, and then SAINT analysis. Over 200 novel interactions were identified and HDAC-centric interactomes revealed HDAC involvement in a variety of processes including ubiquitination and cell cycle regulation. Independent analysis of HDAC11 revealed association with RNA editing and processing, specifically with the SMN subnetwork. Downregulation of HDAC11 was associated with accumulation of splicing defects in ATXN10 gene. The authors complemented their finding with a metabolic labeling approach, I-DIRT, to assess the relative stability of interactions and the binding characteristics of each of the HDACs.

This paper is very impactful in that it provides the first comprehensive and global interactome for HDAC proteins. We agree with the authors that this is particularly important given that HDACs are the targets of several drugs in clinical trials, but the ability to assess their global effect is impaired due to lack of knowledge about the biological pathways and systems in which HDACs are involved. Additionally, the authors have developed a hybrid proteomic approach that can be adapted to the study of other protein interaction networks. By using numerous examples from the literature, they show that the complementarity of the two approaches raises the confidence level of the interactions identified. Overall the paper is well researched, contains a high degree of novelty and is a useful resource platform for further investigations into the HDAC protein family.

Major comments

>The authors should find a way to validate the I-DIRT method. They show binding by two proteins by IP and by colocalization, but they do not show a way to validate their binding strength. This can be done, for example, by salt extraction.

We agree with the reviewer that the use of the integrated I-DIRT/SAINTE approach for determining relative interaction stability is very valuable and can be utilized when studying diverse complexes, and that these findings

could be strengthened by additional experiments. To validate our I-DIRT observations, we have designed and successfully performed two experiments:

1. We performed a biochemical validation of HDAC1 interaction stability. We isolated HDAC1-EGFP using lysis buffer conditions with increasing salt concentrations. To determine the relative resistance of the interactions to the increased salt concentrations, we analyzed the co-isolated proteins by mass spectrometry. Stable interactions, such as the NuRD complex, remained largely associated with HDAC1 at high salt concentration. In contrast, the interactions with transcription factors and zinc finger proteins were depleted in a dose-dependent manner. A clear validation of our I-DIRT results is given by the striking difference between the trends of the zinc finger proteins. The known CtBP complex components, ZNF516 and ZNF217, were stable at high salt concentrations, while the previously uncharacterized zinc finger proteins that I-DIRT predicted as less stable interactions were depleted in a KCl-dependent manner. We illustrate these results in the new Figure 7A.
2. Similarly, we performed additional validation of HDAC7 interaction stability with TBLXR1 and HDAC3. I-DIRT predicted these interactions as less stable, in agreement with the nuclear-cytoplasmic shuttling of HDAC7. Upon isolation of HDAC7-EGFP under lysis buffer conditions with increasing KCl concentrations, we demonstrate by Western blotting that these HDAC7 interactions are diminished in a dose-dependent manner. We illustrate these results in the new Figure 7B.

We have included these new Results on pg 21-22:

“To biochemically validate the ability of the I-DIRT/SAINT approach to determine relative interaction stability, we independently isolated HDAC1-EGFP using lysis buffer conditions with increasing KCl concentrations. We assessed the relative abundance of co-isolated interactions by mass spectrometry. As predicted, stable interactions, such as the NuRD complex, remained largely associated with HDAC1 at high salt concentration (Figure 7A). In contrast, the interactions with transcription factors and zinc finger proteins were depleted in a dose-dependent manner. A clear validation of our I-DIRT results is shown by the striking difference between the relative stability trends of zinc finger proteins. The known CtBP complex components, ZNF516 and ZNF217, were stable at high salt concentrations, while the previously uncharacterized zinc finger proteins that I-DIRT predicted as less stable interactions (Figure 6E-F) were depleted in a KCl-dependent manner (Figure 7A). Similarly, we performed validation of HDAC7 interaction stability with TBLXR1 and HDAC3 (Figure 7B). I-DIRT predicted these interactions as less stable (Figure 6B), in agreement with the nuclear-cytoplasmic shuttling of HDAC7. Upon isolation of HDAC7-EGFP under lysis buffer conditions with increasing KCl concentrations, we observed by Western blotting that TBLXR1 and HDAC3 are diminished in a dose-dependent manner. Altogether, these results support the use of the integrated I-DIRT/SAINT approach for profiling relative protein interaction stabilities.”

>In the HDAC11 analysis only one intron of the ATXN10 gene is shown to be misspliced while the wording of the relevant section claims that SMN1 deletion perturbs the splicing of more than one U12 intron (Boulisfane et al, 2011). Did the authors check any of the other introns? If there wasn't intron retention, was there a difference in splicing efficiency? Can the authors check some other genes with U12 introns to show that this is a general phenomenon?

We took the reviewer's suggestion and have now clarified the impact of HDAC11 on splicing. Analysis of the U12 intron database (<http://genome.crg.es/cgi-bin/u12db/u12db.cgi>) lists that the ATXN10 gene has only one U12-intron (I10), which was previously demonstrated to be mis-spliced in SMN1^{-/-} lymphoblasts (Boulisfane et al., 2011; PMID 21098506). In our experiments, we tested the same intron and demonstrated that there is retention of the I10 intron, but no apparent defect in splicing efficiency (Figure 5E). To determine if this is a broader phenomenon, we performed additional experiments and tested if the U12 intron from Thoc2 (I37) (Boulisfane et al., 2011) was also mis-spliced. Our results show that there are no retention events and there was no difference in splicing efficiency for the Thoc2 gene.

Noteworthy, Boulisfane et al. 2011, report that SMN1^{-/-} lymphoblasts show intron retention defects in only three genes (out of 30 tested), suggesting that this may not be a general phenomenon and could be tissue type specific. This is supported by the report from Zhang et al., 2008, where an examination of alternate splicing changes in the genome reveals that a different set of genes showed splicing changes in the three tissue types tested: brain, kidney and spinal cord.

In summary, our HDAC11 interaction analysis and validation studies provide proof that HDAC11 serves as a part of the SMN1 complex, and establish its role in splicing of the ATXN10 gene. Our results suggest that HDAC11 has a more subtle effect on intron retention than SMN1 deficiency, which may indicate a possible indirect role via the SMN complex or a more specialized role in ATXN10 gene processing. A complete elucidation of these functions

would require many additional studies that are outside the scope of this current manuscript. We have added these new results and the relevant discussion to the new manuscript (pg. 17 and 31).

Pg. 17 - "Homozygous deletion of SMN1 was shown to be accompanied by intron retention in the U12-type intron from the ATXN10 and Thoc2 genes in lymphoblasts derived from patients with spinal muscular atrophy (Boulistfane et al, 2011). We therefore tested the hypothesis that HDAC11 down-regulation would lead to similar splicing defects through disruption of spliceosome function. Upon knockdown of HDAC11 in wild-type CEM-T cells, using qRT-PCR we observed an accumulation of mis-spliced ATXN10 mRNA, which was not detected in the non-targeted siRNA control (Figure 5E). A similar analysis of Thoc2 indicated that there was no splicing defect in the I37 intron from this gene (Figure S6B)."

Pg. 31 - "We demonstrate that HDAC11 downregulation triggers a similar splicing defect of the U12-type intron (I10) from the ATXN10 gene. In contrast, no retention events and no difference in splicing efficiency were observed for the Thoc2 gene, suggesting that HDAC11 has a more subtle effect on intron retention than SMN1 deficiency. HDAC11 may have an indirect role via the SMN complex or a more specialized role in ATXN10 gene processing. These results establish a previously unreported function for HDAC11 in splicing, suggesting a role in the assembly or stabilization of the SMN complex."

Minor comments:

>In section "Protein interaction clustering reflect phylogenetic relationships and functional commonalities among distinct HDACs": The authors write "HDAC bait spectral counts were removed to prevent clustering bias", but the heatmap figure contains HDAC proteins and spectral counts even in the vector representing the experiment in which they should be the bait (e.g. HDAC1 has a yellow color in columns one and two representing the HDAC 1-1 and HDAC1-2 affinity assays). This should be clarified.

In columns 1 and 2, it is HDAC2 that has an intense yellow signal. This is expected as we did not remove spectral counts for HDACs that were found in a different HDAC isolation.

>Can the authors explain why the interactome and heatmap don't contain histone proteins? The interaction table in the Excel file contains histone proteins for several of the HDACs but they are not shown in the heatmap.

The Excel file, Table S2, which lists the SAINT-specific interactions, contains a single histone, histone H1x, which appears in the heatmap and network. The remaining histones listed in Table S3, did not pass the SAINT specificity criteria as the background level of captured histones was relatively high in the EGFP alone isolations. The observation that some abundant and "promiscuous" proteins exhibit non-specific behavior under control conditions is one unfortunate drawback of immunoisolation techniques.

>"Clustering of biological replicates showed highest similarity to each other" - this is not so for HDAC5-1, HDAC5-2, or HDAC4-3. This sentence should be less absolute. Also, why do some HDACs have three biorepeats? Should be clarified.

We have now changed the sentence to, "clustering of biological replicates showed a high degree of similarity to each other" on pg 12. Regarding the additional replicates, these particular HDACs are of great interest for several projects in our lab, and additional independent immunoisolations had been performed by several different lab members, but under identical conditions. Therefore, we decided to include these additional replicates.

>In wording referring to Figure 6, the gene HBP1 (Swanson et al, 2004) is mentioned, but it is not in the figure.

We have labeled the data point corresponding to the relative specificity/stability of HBP1 in Figure 6D and referenced this figure panel in the manuscript on pg 22.

>The reference to Figure S4 should be S3.

Thank you. This has been corrected, and updated to the new numbering, therefore, new Figure S8.

>In Figure 5B, the meaning of the clump of green circles at the bottom is unclear.

We have further clarified this in the figure legend on pg 46, and have included a new supplementary figure showing the complete annotation of the green circles (new Figure S6).

“For clarity, a subset of GO BP term labels relating to detailed RNA metabolic processes were removed (see Figure S6A).”

>In Figure 6C,D the Y axis is named differently. In C it's "SAINT score" and in D it's "SAINT probability". For D, the graph is unnecessary since it contains no points. It's enough to have the color bar with a number legend for the I-DIRT stability ratio and then the chart on the right of the HDACs vs. other proteins.

We have corrected in panel D, SAINT probability to “score” for consistency within the text and other figure panels.

Though we have considered the reviewer’s suggestion for modification of panel D, we ultimately felt that the inclusion of the graph axes unambiguously illustrates an important point, namely that only proteins passing SAINT specificity thresholds were considered for stability profiling. Therefore we have elected to retain the original version of the figure for an increased clarity for a broad audience.

>Figure 2A - where does the metabolic -labeled affinity purification come in to play? The box says LC-MS/MS Analysis...

Both label-free and metabolic-labeled purifications were subjected to peptide analysis by LC-MS/MS, so this step is the same. The most significant differences in the overall workflow were (1) the mixing of heavy-labeled WT cells preceding the affinity purification, as shown in the workflow and Methods section, and (2) isotope-based quantification of peptides, as shown by the I-DIRT graphic of a light peptide and light/heavy peptide pair.

Thank you again for sending us your revised manuscript. We are now satisfied with the modifications made and I am pleased to inform you that your paper has been accepted for publication.

Thank you very much for submitting your work to Molecular Systems Biology.

Reviewer #1 (Remarks to the Author):

I am satisfied with the authors response and their inclusion of additional data and text regarding the EGFP localization and validation of stability predictions with salt elutions. I believe that this is an important paper, will be a valuable addition to the literature, and will be of interest to a wide audience.

Reviewer #2 (Remarks to the Author):

The authors have convincingly addressed the points raised. The manuscript has improved and should be published in MSB.

Reviewer #3 (Remarks to the Author):

The authors have addressed all concerns and revised the manuscript satisfactorily.